# Robust Transfer of Safety-Constrained Reinforcement Learning Agents

**Markel Zubia**[1]**, Thiago D. Simão**[2]**, Nils Jansen**[1,3]
[1]Ruhr Univesity Bochum, Germany
[2]Eindhoven University of Technology, The Netherlands
[3]Radboud University Nijmegen, The Netherlands
`markel.zubia@rub.de, t.simao@tue.nl, n.jansen@rub.de`

## Abstract

Reinforcement learning (RL) often relies on trial and error, which may cause undesirable outcomes. As a result, standard RL is inappropriate for safety-critical applications. To address this issue, one may train a safe agent in a controlled environment (where safety violations are allowed) and then transfer it to the real world (where safety violations may have disastrous consequences). Prior work has made this transfer safe as long as the new environment preserves the safety-related dynamics. However, in most practical applications, differences or shifts in dynamics between the two environments are inevitable, potentially leading to safety violations after the transfer. This work aims to guarantee safety even when the new environment has different (safety-related) dynamics. In other words, we aim to make the process of safe transfer robust. Our methodology (1) robustifies an agent in the controlled environment and (2) provably provides—under mild assumption—a safe transfer to new environments. The empirical evaluation shows that this method yields policies that are robust against changes in dynamics, demonstrating safety after transfer to a new environment.

## 1 Introduction

A prevalent strategy to render reinforcement learning (RL, Sutton & Barto, 2018) safe involves the use of transfer learning, where agents are initially trained in a controlled training environment, such as a simulation or a laboratory (Zhang et al., 2020). Since unsafe interactions with the controlled environment pose no real harm, these agents can be trained via standard safe reinforcement learning methods. The intention is to later transfer these trained agents to target environments where safety may be imperative (García & Fernández, 2015; Peng et al., 2021).

While numerous works have successfully maintained safety after transferring from a controlled training environment to the target (Yang et al., 2023; Feng et al., 2023; Gimelfarb et al., 2021; Karimpanal et al., 2020), their theoretical guarantees of safety rely on the strong assumption that safety-relevant dynamics remain unchanged between the two environments. However, the target environment likely differs from the training environment. Therefore, robustness to changes or shifts in dynamics plays a crucial role in realistic settings involving safety (Meng et al., 2023).

**Problem statement.** This work aims to train an agent in a controlled environment (where safety violations are allowed) to maintain safety throughout the process of transfer-learning, even under the worst-case transition dynamics in the new environment.

Taking inspiration from the method developed by Yang et al. (2023), we focus on a setting where an agent is trained within a reward-free environment, called the *source task* ($\diamond$). Here, the agent learns to navigate the environment safely *without* a reward signal. To account for potential differences in dynamics, we robustify this agent during the training in the source task by adding different kinds of action disturbances. These try to mimic changes in dynamics (different friction, mass, etc.) that would happen in an actual sim-to-real transfer setting, as these changes in dynamics can be viewed as additional disturbances in the system (Başar & Bernhard, 2008). More specifically, we use randomly sampled action noise (Hollenstein et al., 2022), adversarial action noise (Tessler et al.,

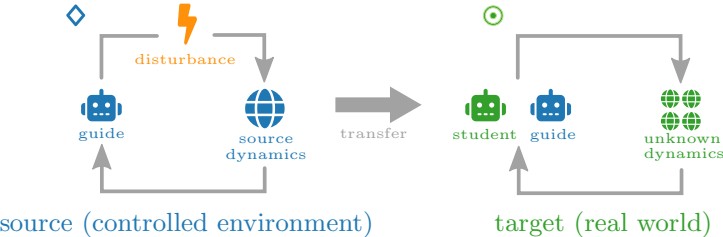

Figure 1: Training the guide under action disturbances in the source task and transferring it to teach the student in a target task with unknown dynamics.

2019), and entropy maximization (Ziebart et al., 2008; Eysenbach & Levine, 2022). Subsequently, this trained agent is transferred to a different environment, called the *target task* ($\odot$), where the reward is revealed. The trained agent, referred to as the *guide*, is then used to train a new agent, the *student*, while making sure that no safety-violations occur during training, as visualized in Figure 1.

**Contributions.** The key contributions of this paper are: 1. Introduce a mathematical framework for modeling robustness in the context of safe transfer learning. 2. Examine and compare how robustness during transfer is affected by entropy maximization, action noise, and adversarial action perturbations. 3. Propose an adaptation of an algorithm for safe transfer-learning developed by Yang et al. (2023) that accounts for uncertainties in the target environment. 4. Empirically and qualitatively analyze the proposed algorithm.

The empirical analysis shows that guides trained with action disturbances tend to have lower worst-case cumulative costs without compromising their overall performance. Furthermore, guides that are robust to changes in dynamics in the source task are better at transferring knowledge in challenging environments, with some achieving completely safe transfers, unlike non-robust ones.

## 2 RELATED WORK

**Safety in RL.** Traditionally, in RL, unsafe behavior is avoided through reward shaping by integrating the safety-related information into the reward signal (Laud & DeJong, 2003). However, such approaches demand significant engineering effort (Kamran et al., 2022; Roy et al., 2022). To avoid reward engineering, safety can be imposed through constraints (García & Fernández, 2015). The literature provides multiple types of constraints (Wachi et al., 2024). For instance, state-wise safety involves establishing state-specific hard constraints (Zhao et al., 2023; Zhan et al., 2024). Shielding is an example of a state-wise safe algorithm, where unsafe actions are blocked at runtime (Alshiekh et al., 2018; Carr et al., 2023). We focus on constraints in expectation, which provides a good trade-off between safety and performance (Altman et al., 2019).

**Robust MDPs.** Robustness is an essential component in sequential decision-making tasks when the transition function is unknown (Suilen et al., 2022; Cubuktepe et al., 2021; Moos et al., 2022; Gu et al., 2025). Robust constrained Markov decision processes (RCMDP) consider a set of potential transition dynamics, called *uncertainty set* (Russel et al., 2021). An agent is robustified in this framework by iteratively exposing it to the worst-case dynamics. Solving RCMDPs may involve computing the worst-case using linear programs (Russel et al., 2021), training an adversary to suggest such dynamics (Bossens, 2024), applying perturbations to the transition function (Goyal & Grand-Clement, 2023), as well as combining homotopy continuation with bisection methods (Behzadian et al., 2021). In contrast to the RCMDP framework, we assume that the uncertainty set is unknown during training to reflect realistic scenarios. We aim to train agents capable of generalization to complete the task in an environment with newly revealed dynamics.

**Adversarial training.** Adversarial training is a common method of achieving robustness (Moos et al., 2022). We may adversarially perturb observations by adding noise to sensors (Liu et al., 2023; Zhang et al., 2021), or the dynamics of the environment, for instance, by applying adversarial

physical forces to the agent (Li et al., 2023). We adopt perturbations to the agent's actions, as preliminary benchmarks showed that they consistently yielded the best results in our setting.

## 3 BACKGROUND

This section covers the theoretical foundations of our method.

**Constrained Markov decision processes.** A constrained Markov decision process (CMDP, Altman, 1999) is characterized by a tuple $\mathcal{M} = (S, A, P, r, c, d, \gamma)$, where $S$ is the continuous state space, $A$ is the continuous action space, $P \colon S \times A \to \text{Distr}(S)$ is the transition probability function with $\text{Distr}(S)$ being the set of probability measures on the Borel sets of $S$, $r \colon S \times A \to \mathbb{R}^+$ is the reward function, $c \colon S \times A \to \mathbb{R}^+$ is the cost function, $d \in \mathbb{R}^+$ is the cost-budget, and $\gamma \in [0, 1)$ is the discount factor. A *stationary stochastic policy* is a function $\pi \colon S \to \text{Distr}(A)$ that suggests a possibly stochastic action given the current state. We define the *set of stationary stochastic policies* as $\Pi$. The *expected cumulative reward* of policy $\pi$ in state $s$ over a finite horizon of length $T$ is $V^\pi(s) = \mathbb{E}_\pi \left[ \sum_{t=0}^T r(s_t, a_t)\gamma^t \Big| s_0 = s \right]$, and the *expected cumulative cost* is defined similarly as $C^\pi(s) = \mathbb{E}_\pi \left[ \sum_{t=0}^T c(s_t, a_t)\gamma^t \Big| s_0 = s \right]$. The goal of the CMDP framework is to find an optimal policy $\pi^*$, that is, a policy that maximizes the expected cumulative reward while keeping the expected cumulative cost below $d$ for all states $s \in S$:

$$\max_{\pi \in \Pi} V^\pi(s) \quad \text{s.t.} \quad C^\pi(s) \leq d.$$

The CMDP objective can be transformed with the Lagrangian relaxation method (Bertsekas, 1997),

$$\max_{\pi \in \Pi} \min_{\lambda \geq 0} V^\pi(s) - \lambda(C^\pi(s) - d).$$

We also define the *expected return* as $Q_\pi^r(s, a) = \mathbb{E}_\pi \left[ \sum_{t=0}^T \gamma^t r(s_t, a_t) \Big| s_0 = s, a_0 = a \right]$, and the *expected cost-return*, $Q_\pi^c$, is defined analogously.

**Safe transfer-learning.** In the transfer learning framework for RL, an agent must leverage the knowledge gained from the *source task* ($\diamond$) to learn the *target task* ($\odot$) more efficiently (Zhu et al., 2023). Said tasks are modeled by CMDPs in safety-critical applications, where the constraints model aspects concerning safety (García & Fernández, 2015; Peng et al., 2021). Yang et al. (2023) propose a reward-free source task $\mathcal{M}^\diamond = (S^\diamond, A^\diamond, P^\diamond, \emptyset, c^\diamond, d^\diamond, \gamma)$, where one can train an agent called the *guide*, with policy $\pi^\diamond$, that can safely navigate it without relying on a reward signal. This trained guide can then be used to train the *student* agent, with policy $\pi^\odot$, in the target task $\mathcal{M}^\odot = (S^\odot, A^\odot, P^\odot, r^\odot, c^\odot, d^\odot, \gamma)$.

Various transfer metrics exist for evaluating the extent to which one agent benefits from leveraging the knowledge of another agent (Taylor & Stone, 2009; Yang et al., 2023). One such metric is *safety jump-start*, which measures the difference in cumulative costs during the initial epoch between an agent that uses prior knowledge and one learning from scratch. Similarly, $\Delta$ *time to safety* assesses the difference in time required to reach safe behavior. Analogously, *return jump-start* and $\Delta$ *time to optimum* concern the reward signal.

## 4 ROBUST SAFE TRANSFER-LEARNING FRAMEWORK

This paper focuses on training a guide in the source task and transferring it to a target task that has different dynamics, which are unknown during the training of the guide, to then train a student while avoiding safety violations. Our source task is modeled by a reward-free CMDP $\mathcal{M}^\diamond = (S^\diamond, A^\diamond, P^\diamond, \emptyset, c^\diamond, d^\diamond, \gamma)$ as already introduced in the previous section. Since the dynamics of the target task are not known, we choose a more robust model. In particular, we model the target task as a CMDP where the transition function has been replaced by a *set of possible transition dynamics functions*, $\mathcal{M}^\odot = (S^\odot, A^\odot, U^\odot, r^\odot, c^\odot, d^\odot, \gamma)$, where every $P^\odot \in U^\odot$ is a dynamics function $P^\odot \colon S^\odot \times A^\odot \to \text{Distr}(S^\odot)$. We refer to $U^\odot$ as the *uncertainty set*. One possible instance of such an uncertainty set is to define intervals of probabilities on transitions, creating infinitely many potential dynamics of the target task.

### 4.1 Safety guarantees in worst-case environments

Since the intention is to transfer the policy of the guide, $\pi^\diamond$, from $\mathcal{M}^\diamond$ to $\mathcal{M}^\odot$, both environments should share the same action space:

**Assumption 1.** $A^\diamond = A^\odot = A$.

Additionally, notice that the guide can only process observations from the source task's state space $S^\diamond$. Thus, to transfer the guide to the target task, we need to be able to map the state space of the target task to the source space of the target task. In particular, we are interested in cases where this mapping preserves information about the dynamics concerning safety to guarantee a safe transfer. This concept is introduced by Li et al. (2006):

**Definition 1.** *Function* $\sigma \colon S \to S^\dagger$ *is a* $Q^c$*-irrelevant abstraction for* $\mathcal{M} = (S, A, P, r, c, d, \gamma)$ *whenever* $\forall s, s' \in S, \forall a \in A, \forall \pi \in \Pi$, $\sigma(s) = \sigma(s') \Rightarrow Q^c_{\pi,\mathcal{M}}(s, a) = Q^c_{\pi,\mathcal{M}}(s', a)$.

Similar model-invariant abstractions have been used in the CMDP setting to guarantee that safety constraints are satisfied (Simão et al., 2021). Now, we assume that such a mapping exists from $S^\odot$ to $S^\diamond$, and that this function preserves boundedness:

**Assumption 2.** *There is a* $Q^c$*-irrelevant abstraction* $\sigma \colon S^\odot \to S^\diamond$ *for* $\mathcal{M}^\odot$ *that preserves boundedness.*

Given an arbitrary $Q^c$-irrelevant abstraction, we consider how this function can map any CMDP $\mathcal{M}$ to a new CMDP $\mathcal{M}^\dagger$, the latter being referred to as the *abstracted task* (Li et al., 2006). Refer to Appendix C for an explicit construction.

To clarify, even though $\sigma$ maps the states in the target task to states in the source task, it does not imply that the dynamics in the two tasks are similar, and it in fact does not concern the source task's dynamics in any manner. More precisely, Assumption 2 implies that the states in the target task can be mapped to the states in the source task, where the information about the *target task's dynamics* is preserved, but the *source task's dynamics* are not referenced and could thus be vastly different from those of the target task.

Intuitively, $\sigma$ strips away information that is not safety-relevant, to return a new abstracted CMDP where the dynamics concerning safety are the same as in the initial CMDP. Therefore, it has been formally proven that an agent that is safe in the initial task is also safe in the abstracted task:

**Lemma 1.** *(Abel et al., 2016) Let* $\sigma \colon S \to S^\dagger$ *be* $Q^c$*-irrelevant, and tasks* $\mathcal{M} = (S, A, P, r, c, d, \gamma)$ *and* $\mathcal{M}^\dagger = (S^\dagger, A, P^\dagger, r^\dagger, c^\dagger, d, \gamma)$ *constructed as in Appendix C. Then,* $\forall s \in S, \forall a \in A, \forall \pi \in \Pi^\dagger$, $Q^c_{\pi,\mathcal{M}^\dagger}(\sigma(s), a) = Q^c_{\pi \circ \sigma, \mathcal{M}}(s, a)$. $\qquad\square$

We have not yet imposed any restrictions on the uncertainty set, which means that the various dynamics transition functions in this set could potentially be infinitely different from each other. Therefore, it is reasonable to disallow this by supposing that the uncertainty set is bounded:

**Assumption 3.** $U^\odot$ *is bounded, that is,* $\exists \varepsilon \in \mathbb{R}, \forall P, P' \in U^\odot$, $\|P - P'\| \leq \varepsilon$.

In the following theorem, we prove that an agent that is robust against changes in dynamics in the source task will retain part of this robustness when transferred to the target task:

**Theorem 1.** *Given Assumptions 1, 2, and 3, if* $d^\diamond \leq d^\odot$ *then there exists* $\delta \in \mathbb{R}$ *such that if policy* $\pi$ *is safe in* $\mathcal{M}^\diamond_P$ *for all* $P$ *where* $\|P - P^\diamond\| \leq \delta$, *then* $\pi \circ \sigma$ *is safe in* $\mathcal{M}^\odot_{P'}$ *for all* $P' \in U^\odot$.

*Proof.* Since $U^\odot$ is bounded (Assumption 3) and $\sigma$ preserves boundedness (Assumption 2), there is

$$\delta = \max_{P \in U^\odot} \|P^\dagger - P^\diamond\|,$$

where the abstracted dynamics $P^\dagger$ is constructed as in Appendix C. Now, suppose that $\pi$ is safe for all $P$ with $\|P - P^\diamond\| \leq \delta$, that is, $Q^{c,\diamond}_{\pi,P}(s, a) \leq d^\diamond$. Then, for all $P \in U^\odot$, we have $Q^{c,\diamond}_{\pi,P^\dagger}(s, a) \leq d^\diamond$ by construction of $\delta$. Thus, $Q^{c,\odot}_{\pi \circ \sigma, P}(s, a) \leq d^\diamond \leq d^\odot$ given Lemma 1 and premise $d^\diamond \leq d^\odot$. $\qquad\square$

This proof is visualized in Figure 2. Theorem 1 justifies increasing $\delta$ in the source task with robustification methods before transferring to a target task with uncertain dynamics, which is an essential component in the method we introduce in Section 5.

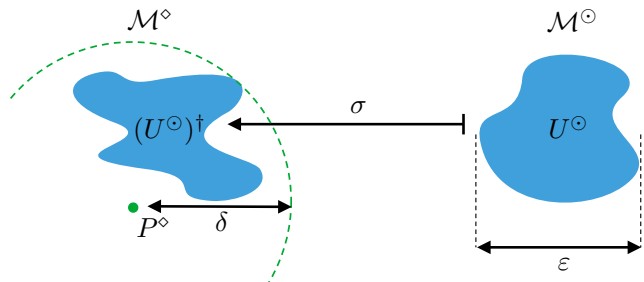

Figure 2: Visualization of the space of dynamics functions from Theorem 1.

## 4.2 ROBUST TRANSFER METRICS

**Key insight.** While an agent that is robust in the source task will also be robust in the target task, as shown in Theorem 1, this is not enough to be able to perform the transfer of knowledge successfully. To illustrate, notice that in many environments the "do nothing" policy is safe everywhere in $U^{\odot}$, even though it is clear that this policy provides no value to train an agent in the target task. Thus, sometimes an agent that prioritizes exploration while taking more risks may be more useful for training another agent in the target task, compared to an overly conservative one, resulting in an overall lower cumulative cost. This serves as motivation for introducing metrics that measure the usefulness of the guide's knowledge from the perspective of safety and robustness.

Let's say that algorithm $\Phi$ uses the knowledge from the agent trained in the source task, $\pi^{\diamond}$, to train a new agent $\pi^{\odot}$ in the target task. To do so, the algorithm needs to sample $K$ trajectories of length $L$ in $\mathcal{M}^{\odot}$, generating a history of environment interactions $\{\rho_i\}_{i=1...,K}$. Since the algorithm may be nondeterministic, we have a history of probability distributions over the space of trajectories, denoted by $\rho_i \in \text{Distr}((S \times A)^* \times S)$ where the Kleene operator $(\cdot)^*$ indicates that we may have finite but arbitrarily long sequences of state and action pairs. This is expressed compactly as $\Phi(\pi^{\diamond}, \mathcal{M}^{\odot}) = \{\rho_i\}_{i=1...,K}$, and we denote learning from scratch as $\Phi(\emptyset, \mathcal{M}^{\odot})$.

Various transfer metrics exist for evaluating to which extent the target policy $\pi^{\odot}$ benefits from leveraging the knowledge of $\pi^{\diamond}$ (Taylor & Stone, 2009), some of which have been adapted to account for safety by Yang et al. (2023). As these metrics have only been defined informally in prior work, we next provide rigorous definitions.

**Definition 2.** *Given* $\Phi(\pi^{\diamond}, \mathcal{M}^{\odot}) = \{\rho_i\}$ *and* $\Phi(\emptyset, \mathcal{M}^{\odot}) = \{\rho'_i\}$, *the* safety jump-start *is defined as*

$$\mathfrak{J}(\pi^{\diamond}, \mathcal{M}^{\odot}) = \mathbb{E}\left[\frac{\mathfrak{c}(\tau) - \mathfrak{c}(\tau')}{\mathfrak{c}(\tau')} \,\middle|\, \tau \sim \rho_1, \tau' \sim \rho'_1\right],$$

*where* $\mathfrak{c}((s_1, a_1, \ldots, s_{L+1})) = \max(d^{\odot}, \sum_{i=1}^{L} \gamma^i c(s_i, a_i))$.

In other words, the safety jump-start measures the difference in excess cumulative costs during the initial trajectory or epoch between an agent that uses prior knowledge and one learning from scratch.

**Definition 3.** *The* $\Delta$ time to safety *assesses the difference in time required to reach safe behavior. Given* $\Phi(\pi^{\diamond}, \mathcal{M}^{\odot}) = \{\rho_i\}$ *and* $\Phi(\emptyset, \mathcal{M}^{\odot}) = \{\rho'_i\}$, *the metric is defined as*

$$\mathfrak{C}(P, \pi^{\diamond}) = \mathbb{E}\left[\mathfrak{m}(\{\tau'_i\}) - \mathfrak{m}(\{\tau_i\}) \mid \tau_i \sim \rho_i, \tau'_i \sim \rho'_i\right],$$

*where* $\mathfrak{m}(\{\tau_i\}) = \min_{t=1,\ldots,L} t$ *such that* $\forall t' \geq t, \mathfrak{c}(\tau_{t'}) \leq d^{\odot}$.

**Definition 4.** *Analogously to the safety jump-start and* $\Delta$ *time to safety, the* return jump-start *and* $\Delta$ time to optimum *concern the reward signal, respectively denoted by* $\mathfrak{B}(\mathcal{M}^{\odot}, \pi^{\diamond})$ *and* $\mathfrak{R}(\mathcal{M}^{\odot}, \pi^{\diamond})$.

The transfer-learning problem at hand is to train the guide in a reward-free source task modeled by a CMDP, $\mathcal{M}^{\diamond}$, to later use it to learn a student policy in the target task modeled by a CMDP, $\mathcal{M}^{\odot}$, with uncertainty set $U^{\odot}$. It is important to bear in mind that $\mathcal{M}^{\odot}$ is completely inaccessible during the training of the guide in $\mathcal{M}^{\diamond}$, meaning that the guide cannot interact with $U^{\odot}$ during training.

**Formal statement of the problem.** Given a reward-free source CMDP $\mathcal{M}^{\diamond}$ and a target CMDP $\mathcal{M}^{\odot}$ with uncertainty set $U^{\odot}$, minimize the worst-case safety jump-start while keeping $\Delta$ time to

optimum above a threshold $d_r$:

$$\min_{\pi \in \Pi} \max_{P \in U^\odot} \mathfrak{J}(P, \pi) \text{ s.t. } \mathfrak{R}(P, \pi) \geq d_r.$$

Even though the problem statement only includes the safety jump-start and $\Delta$ time to optimum of the guide, we are interested in measuring the $\Delta$ time to safety and return jump-start as well.

**Definition 5.** *Given a policy $\pi \in \Pi$, the* worst-case *transition dynamics is*

$$P^+ = \arg\max_{P \in U^\odot} \mathfrak{J}(P, \pi).$$

In practice, estimating the worst-case transition dynamics may require evaluating a policy within various $P \in U^\odot$ and picking the environment that yields the highest cost. Consequently, notice that we are likely to choose outliers through this process, causing the approximate worst-case safety jump-start to have high variance. In an attempt to mitigate this issue, rather than selecting only one environment, one can pick a portion $p \in [0, 1]$ of the environments where the agent performs the worst. The following definition formalizes this concept.

**Definition 6.** *For $p \in [0, 1]$, the* $p$-tail *of the safety jump-start is*

$$\mathfrak{J}^{\leq p}(\mathcal{M}^\odot, \pi) = \int_B \mathfrak{J}(\mathcal{M}_P^\odot, \pi) \, dP,$$

*if there exists $y \in \mathbb{R}$ such that $\frac{\|B\|}{\|U^\odot\|} = p$, where $B = \{P \in U^\odot \mid \mathfrak{J}(\mathcal{M}_P^\odot, \pi) \geq y\}$.*

Additionally, $\mathfrak{J}^{\leq 1}(\mathcal{M}^\odot, \pi)$ is the *average performance* of $\pi$ within the entirety of $U^\odot$. While the problem statement does not include the $p$-tail and average performance, we will keep track of them in the empirical analysis.

## 5 ROBUST GUIDED SAFE EXPLORATION

Our method trains a robust guide via action disturbances in the source task and transfers it to the target task to train the student safely.

### 5.1 TRAINING THE GUIDE

Since the source task is a reward-free CMDP, $\mathcal{M} = (S, A, P, \emptyset, c, d, \gamma)$, it is likely that an agent trained in this environment would learn to "do nothing", as it would only need to satisfy the safety constraints. Therefore, we use a distance bonus to encourage the guide to explore the environment,

$$r^b(s_t, a_t) = \mathbb{E}\left[\|s_t^\ddagger - s_{t+1}^\ddagger\| \,\Big|\, s_{t+1} \sim P(s_t, a_t)\right],$$

where $\|\cdot\| \colon S^\ddagger \to \mathbb{R}$ is a norm and $(\cdot)^\ddagger$ is a state abstraction function (Yang et al., 2023).

To transfer the guide to the uncertain target task, we must first robustify it during its training in the source task. We employ three domain-agnostic robustification techniques.

**Entropy maximization.** To maximize the entropy of the agent, we include the following term in the reward signal $r_t^\diamond(s_t, a_t) = r_t^b(s_t, a_t) + \alpha r_t^\mathcal{H}(s_t, a_t)$, where $r_t^\mathcal{H}(s_t, a_t) = \log \frac{1}{\pi(a_t|s_t)}$.

**Random action noise.** Instead of sampling the action directly from the agent, some random noise is added: $a \sim (1 - \alpha)\pi(s_t) + \alpha\mathcal{N}(\mu, \sigma)$.

**Adversarial perturbations.** The guide is trained in a noisy action robust MDP (Tessler et al., 2019): $a \sim (1 - \alpha)\pi(s_t) + \alpha\bar{\pi}(s_t)$, where the guide is trying to maximize the objective $r_t^\odot = r_t^b$ while satisfying the cost constraints, and the adversary is trying to maximize the cumulative cost.

Prior work has established that perturbation size ($\alpha$) provably makes the policy robust to some $\delta$ (Feng et al., 2020; Eysenbach & Levine, 2022). Therefore, as we increase the value of $\alpha$, we are effectively computing a robust policy for a larger uncertainty set in the source task. Consequently, the agent will be safe in a larger set of target tasks. As a theoretical bound between $\alpha$

and $\delta$ can only be shown in specific scenarios, we adopt an empirical approach in the next section and evaluate the robustness effect across different values of $\alpha$.

Note that Theorem 1 does not state that there always is a robustly safe policy. If $\delta$ is too large, such a policy may not exist. In practical terms, if we fail to find a feasible policy for a given $\delta$, our method still attempts to find a conservative policy in the source target, which ensures a reduction of safety violations in the target environment. Alternatively, we could search for the largest $\delta' < \delta$ such that a feasible policy exists. Nevertheless, this policy can easily be found in the constrained RL setting. For instance, in problems exhibiting an action with zero cost, such as in the environments used in the empirical evaluation, the agent can choose this action, which ensures it will satisfy the constraints.

## 5.2 TRANSFERRING KNOWLEDGE TO THE STUDENT

To transfer the knowledge of the guide to the student, our approach is similar to (Yang et al., 2023). We want to ensure safety throughout by sampling from the student whenever the cost signal is zero, and otherwise we sample from the guide:

$$\pi^b(s_t) = \begin{cases} (\pi^\diamond \circ \sigma)(s_t) & \text{if there is a } t' \leq t \text{ where } c_{t'} > 0, \\ \pi^\odot(s_t) & \text{else.} \end{cases}$$

It is desirable to let the student's policy imitate the guide's whenever the cumulative cost is above the safety threshold, while this is not necessary once the student behaves safely. We achieve this effect by adding a new term $r_t^{\sim}$ to the reward signal measuring the similarity between the two policies, weighted by the Lagrangian multiplier, $\lambda$. When both policies are nondeterministic, $r_t^{\sim}$ measures the KL divergence between the two distributions. Otherwise, if either policy is deterministic, we compare the deterministic action with the mean of the nondeterministic policy's distribution.

## 6 EMPIRICAL ANALYSIS

We train one guide for each method of robustification (Section 5.1). Our empirical analysis aims to answer the following questions:

1. How does each guide perform in the source task?
2. How does each guide perform in the target task's worst-case dynamics?
3. What is the safety jump-start of each guide in the target task?
4. How is the full training of the student affected by the guide's robustness?

We evaluate our method [1] on benchmark environments created using a framework for safe reinforcement learning called *Safety-Gymnasium* (Ji et al., 2023). In these environments, the RL agent controls a robot that must reach the goal while avoiding the hazards. Appendix B provides more details.

**The source tasks.** There are three source tasks: ($\mathcal{M}_1^\diamond$, $\mathcal{M}_2^\diamond$, and $\mathcal{M}_3^\diamond$). These environments may contain static obstacles called *hazards* and dynamic obstacles called *vases*, which are always constrained. Additionally, all three environments have constrained walls to discourage the agent from going out of bounds. The environments differ as follows: $\mathcal{M}_1^\diamond$ has 1 hazard located in the center; $\mathcal{M}_2^\diamond$ has 5 hazards that change location every epoch, and $\mathcal{M}_3^\diamond$ has 8 hazards and 8 vases that change location every epoch.

**The target tasks.** The target environments $\mathcal{M}_1^\odot$, $\mathcal{M}_2^\odot$, and $\mathcal{M}_3^\odot$ are similar to their respective source environments $\mathcal{M}_1^\diamond$, $\mathcal{M}_2^\diamond$, and $\mathcal{M}_3^\diamond$, except for two major differences.

1. The task in the target environments is to reach a specific location, called the *goal*. Therefore, the observations in the target tasks have the additional measurements that concern the reward signal: a LIDAR in $[0, 1]^{16}$ for detecting the goal. We have a mapping $\sigma \colon S^\odot \to S^\diamond$ which simply strips away the information concerning the reward signal. It is easy to show that $\sigma$ is a $Q^c$ irrelevance abstraction, satisfying Assumption 2.

---

[1] The source code is available on `https://github.com/ai-fm/safe-and-robust-transfer`

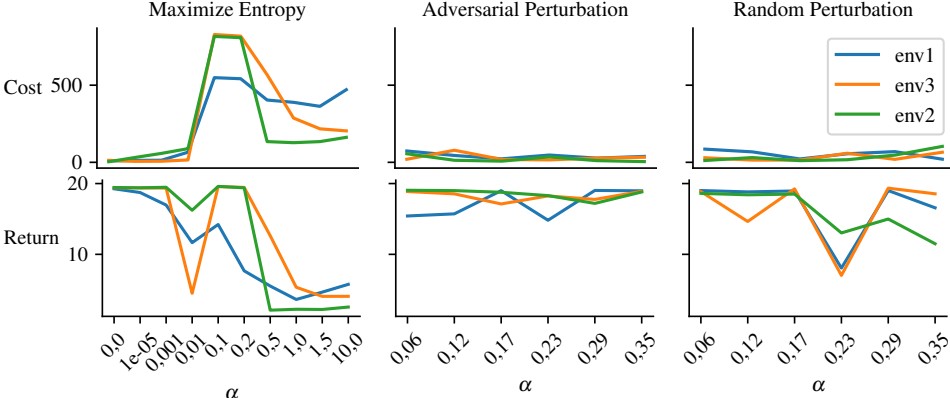

Figure 3: The cost and return in the last epoch of each type of guide.

2. The transition dynamics function in the target tasks is uncertain. In our practical implementation, this is achieved by modifying the physics parameters concerning the *friction* and *mass* of the agent in the simulated environments. The friction refers to the force that opposes the motion of the robot, and a higher friction value causes the agent to decelerate quicker. The mass of the agent affects its moment of inertia, meaning that it impacts how quickly the agent can change its velocity.

One can define the function $u \colon \mathbb{R}^2 \to (S^{\odot} \times A \to \mathrm{Distr}(S^{\odot}))$ that maps the friction and mass of the agent to the corresponding transition dynamics function. Thus, notice that $u$ preserves boundedness, which implies that $U^{\odot}$ is bounded, satisfying Assumption 3.

## 6.1 PERFORMANCE OF THE GUIDES IN THE SOURCE TASK

We train various guides with the three robustification techniques: random action noise, adversarial action perturbations, and entropy maximization. The guides are trained with different weights of the robustification methods to see which values of these parameters yield the best outcomes. The ranges of these weights are based on the results of prior work (Tessler et al., 2019; Liu et al., 2023): the random noise is weighted by $\alpha \in \{0.06, 0.12, 0.17, 0.23, 0.29, 0.35\}$; the adversarial perturbation weights are $\alpha \in \{0.06, 0.12, 0.17, 0.23, 0.29, 0.35\}$; and the entropy bonus is weighted by $\alpha \in \{10^{-5}, 10^{-3}, 10^{-2}, 0.1, 0.2, 0.5, 1.0, 1.5, 10.0\}$.

Figure 3 shows the cumulative returns and costs at the very last epoch of the three kinds of agents trained with the different values of the robustification weights. Most guides learn a safe policy that obtains a cumulative reward slightly below 20, except for the entropy-maximizing agent when $\alpha \geq 0.1$, which does not learn a proper policy, presumably due to the overly high entropy.

## 6.2 GUIDES' PERFORMANCE IN THE TARGET TASK'S WORST-CASE DYNAMICS

The uncertainty set $(U^{\odot})$ consists of uncountably many dynamics transition functions, parameterized by the mass $(m)$ and friction $(\eta)$ of the agent, making it challenging to compute the worst-case dynamics. Therefore, we restrict the uncertainty set to a finite subset $(\bar{U}^{\odot})$ by discretizing the values of the parameters to $m = m_1, \ldots, m_N$, and $\eta = \eta_1, \ldots, \eta_N$. In our experiments, we use $N = 8$ values for each parameter by letting $m_i = (0.5 + \frac{i-1}{7})m^{\diamond}$ and $\eta_i = (0.5 + \frac{i-1}{7})\eta^{\diamond}$ for $i = 1, \ldots, 8$, where $m^{\diamond}$ and $\eta^{\diamond}$ correspond to the dynamics in the source task.

First, we evaluate all guides within their respective source tasks, but instead of using the source task's dynamics $(P^{\diamond})$, we evaluate them within the discretized uncertainty set of the target task, $\bar{U}^{\odot}$. This experiment provides insights into the relationship between the different robustification algorithms and the robustness they provide. Figure 4 shows the cumulative costs of a robust and non-robust agent evaluated in the dynamics functions of the discretized uncertainty set.

The robust guide can navigate the source task safely even when the dynamics are unfavorable, while the non-robust agent struggles to maintain a safe expected cost when the shift in dynamics is large.

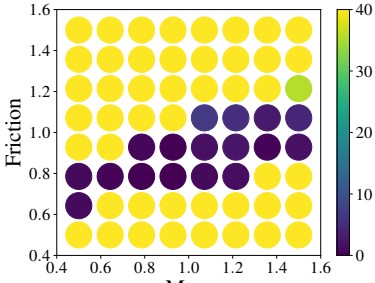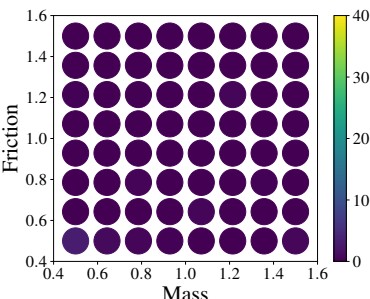

Figure 4: The expected cumulative costs in $\mathcal{M}_1^\diamond$ of a guide trained with no action noise (left) and a guide with an action noise of $\alpha = 0.29$, each dot representing a different dynamics function, with mass on the $x$-axis and friction on the $y$-axis.

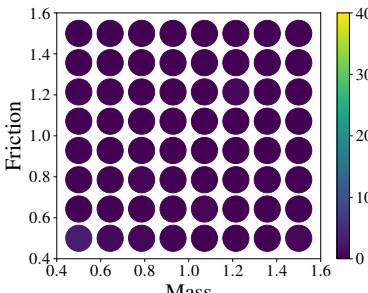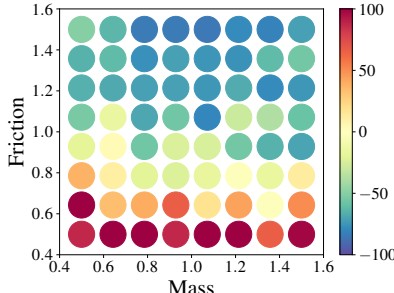

Figure 5: Expected cumulative cost in the source task (left) and the safety jump-start in the target task as a percentage (right) of the agent trained with $\alpha = 0.29$ within $\mathcal{M}_1$. Low values of the safety jump-start are desirable, where a jump-start of $-100$ concerns a fully safe transfer.

Heatmaps like the ones shown in Figure 4 have been computed for all three robustification methods and different values of $\alpha$, and they can be seen in Appendix F. Once these heatmaps are calculated, it is easy to obtain the worst-case expected cost, as well as the $p$-tail and the average, shown in Tables 2, 3, and 4. It is crucial to note that the values shown in the tables are the expected costs of one batch of guides, implying that conclusions cannot be drawn from the comparison the values of $\alpha$ during training with their respective costs, due to a lack of statistical significance. Nevertheless, since the expected costs are computed quite accurately, it is correct to claim that some guides are more robust than others, which will be useful in the next sections to measure how the robustness of the guide impacts the student's training in the target task.

### 6.3 SAFETY JUMP-START OF EACH GUIDE IN THE TARGET TASK'S WORST-CASE DYNAMICS

The safety jump-start assesses the difference in cumulative costs during the first epoch between the student learning from a guide and one learning from scratch. Measuring the safety jump-start is computationally cheap compared to other metrics, as it requires training the student for only one epoch as opposed to fully training said student.

We transfer the knowledge of the guides to two kinds of students: a nondeterministic student (SAC) and a deterministic one (DDPG). One major problem with transferring the knowledge to a deterministic student is that it is much likelier to end up in unrecoverable states by selecting the same actions over and over, rendering the control-switch rescue method completely ineffective. To lessen this issue, the deterministic student is trained with a relatively high random action noise of $\alpha = 0.75$.

Appendix G shows tables collecting the worst-case, $0.1$-tail, and average safety jump-starts measured from every guide to the two kinds of students. An interesting observation when comparing these with Tables 2, 3, and 4, is that policies that appear robust in the source task may not necessarily have the ability to effectively transfer their knowledge to a student in their target task. An extreme

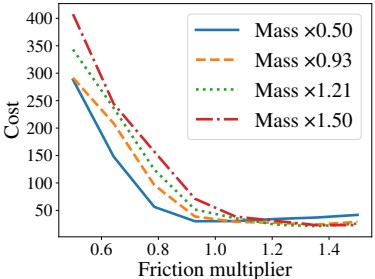

Figure 6: The expected cumulative costs in $\mathcal{M}_2^{\odot}$ for the average guide given each mass-friction pair.

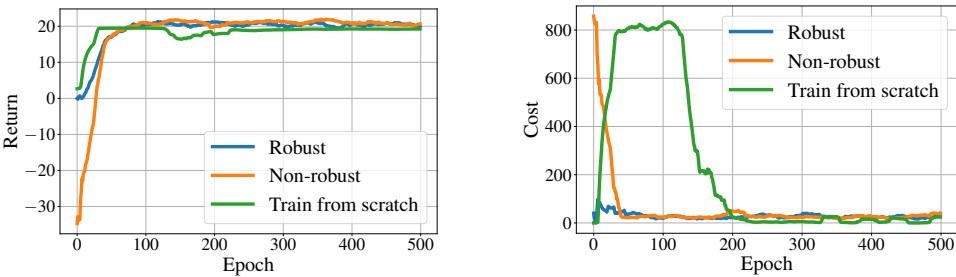

Figure 7: Comparison of the cumulative rewards and costs during training between the students trained with the robust guide ($\alpha = 0.29$) and the non-robust guide (Yang et al., 2023).

example of such a phenomenon can be seen in Figure 5, where an agent that is robust in the source task has an exceptionally poor safety jump-start in worst-case environments.

### 6.4 EFFECT OF THE GUIDE'S ROBUSTNESS ON THE STUDENT'S TRAINING

To determine how the transfer is affected by the robustness of the guide, we train the student with a robust guide and a non-robust one. For the robust guide, we will use the agent trained with action noise in $\mathcal{M}_2$ where $\alpha = 0.29$. The non-robust guide is trained within $\mathcal{M}_2$ using the method introduced by Yang et al. (2023). We then transfer the guides to the dynamics that appear most challenging based on the chart shown in Figure 6: $m^{\odot} = 1.5m^{\diamond}$ and $\eta^{\odot} = 0.5\eta^{\diamond}$.

The reward and cost during training of the students trained with robust and non-robust guides are shown in Figure 7. Since the goal of the transfer is to avoid safety violations within the target task, the transfer done with the robust guide seems ideal. Even though the training takes place in a task with unfavorable dynamics, the safety jump-start with the robust guide nears -100%, meaning that the behavior policy is safe from the very start. Moreover, the return jump-start of the student with the robust guide is much greater than that of the student with a non-robust guide.

## 7 CONCLUSIONS

We propose a method to transfer agents to environments with different or even worst-case dynamics while satisfying safety constraints. The empirical evaluation shows that, in general, agents trained with action disturbances have lower worst-case expected cumulative costs without sacrificing the expected return. In addition, we observe that guides that are robust in the source task do not always have a favorable worst-case jump-start, which backs our theoretical insights. Lastly, the agents with low worst-case safety jump-start demonstrate better capability for transferring knowledge to the student in unfavorable dynamics, compared to non-robust guides, where some robustified agents have been shown capable of achieving a fully safe transfer.

ACKNOWLEDGMENTS

We would like to thank the anonymous reviewers for their valuable feedback. This research has been partially funded by the ERC Starting Grant 101077178 (DEUCE).

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

# A    HYPERPARAMETERS

The hyperparameters in our method are summarized in Table 1. All actor and critic networks are modeled by a multilayer perceptron (MLP).

| Parameter | $\mathcal{M}_1$ | $\mathcal{M}_2$ | $\mathcal{M}_3$ |
|---|---|---|---|
| Actor network size | [256, 256] | [256, 256] | [256, 256] |
| Critic network size | [256, 256] | [256, 256] | [256, 256] |
| Size of replay buffer | $10^6$ | $10^6$ | $10^6$ |
| Batch size | 256 | 256 | 256 |
| Steps per epoch | 2000 | 2000 | 2000 |
| Number of epochs | $10^6$ | $10^6$ | $10^6$ |
| Actor learning rate | $5 \cdot 10^{-6}$ | $5 \cdot 10^{-6}$ | $5 \cdot 10^{-6}$ |
| Critic learning rate | $10^{-3}$ | $10^{-3}$ | $10^{-3}$ |
| Lambda learning rate | $5 \cdot 10^{-7}$ | $5 \cdot 10^{-7}$ | $5 \cdot 10^{-7}$ |
| Safety constraint | 5 | 8 | 25 |

Table 1: The hyperparameters used in the experiments.

# B    THE ENVIRONMENTS

The actor is a small robot called the *point* that can move forwards, backwards, and steer left-to-right. The action space is $[-1, 1]^2$, where the first value is for the throttle and the second one is for steering. The observation space has the following components:

- Pseudo-LIDAR for hazards in $[0, 1]^{16}$.
- Pseudo-LIDAR for vases in $[0, 1]^{16}$.
- Velocimeter in $(-\infty, \infty)^3$.
- Accelerometer in $(-\infty, \infty)^3$.
- Gyroscope in $(-\infty, \infty)^3$.
- Magnetometer in $(-\infty, \infty)^3$.

The pseudo-LIDAR casts 16 rays in different angles. It is termed "pseudo" because, unlike with real LIDARs, the rays can go through objects. The other sensors measure in three dimensions: a velocimeter for velocity (m/s), an accelerometer for acceleration (m/s$^2$), a gyroscope for angular velocity (rad/s), and a magnetometer for magnetic flux (Wb).

Renders of the tasks are shown in Figures 8, 9, and 10.

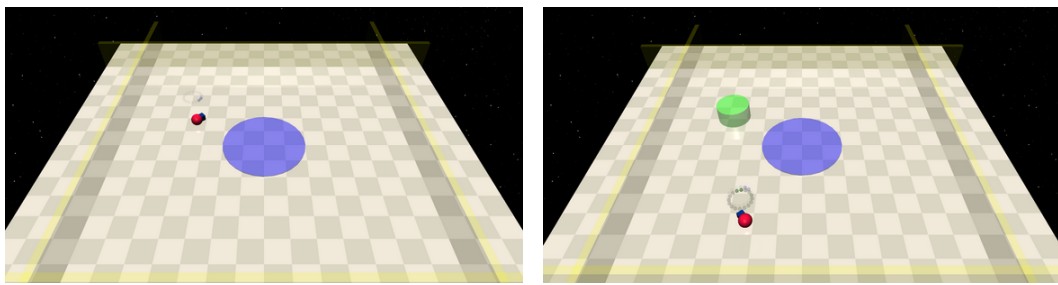

Figure 8: Renders of tasks $\mathcal{M}_1^\diamond$ (left) and $\mathcal{M}_1^\odot$ (right).

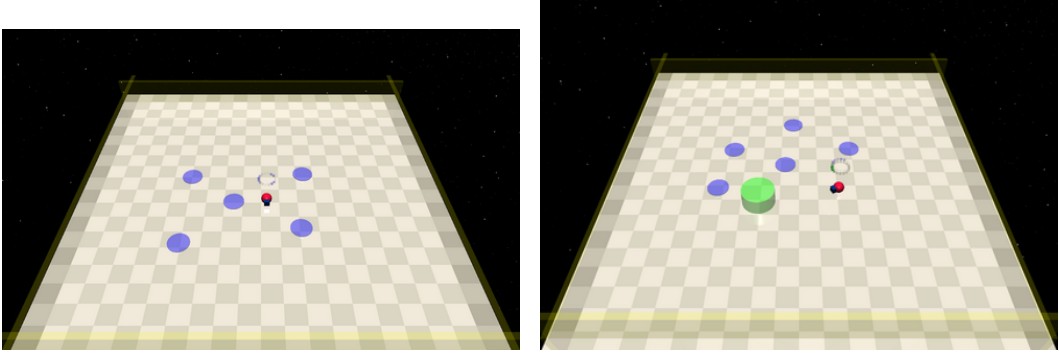

Figure 9: Renders of tasks $\mathcal{M}_2^{\diamond}$ (left) and $\mathcal{M}_2^{\odot}$ (right).

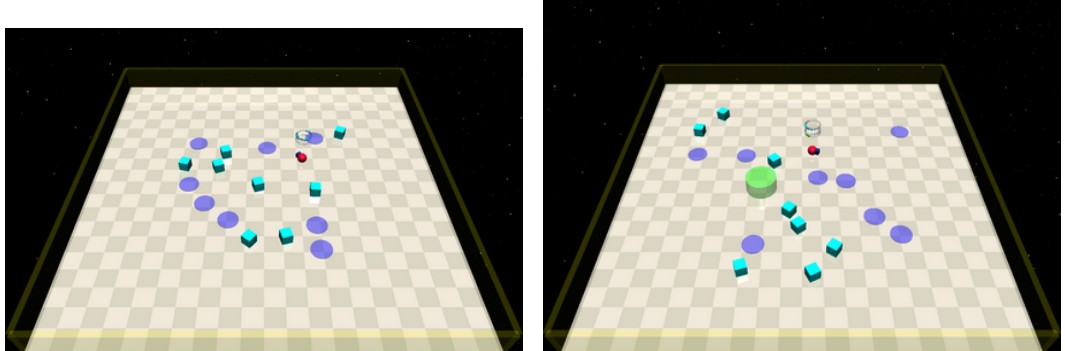

Figure 10: Renders of tasks $\mathcal{M}_3^{\diamond}$ (left) and $\mathcal{M}_3^{\odot}$ (right).

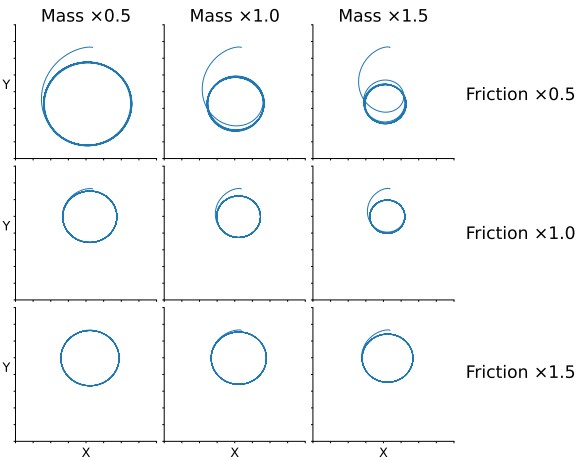

Figure 11: The trajectories of $\pi(\_) = (1.0, 1.0)$ where the mass and damping are multiplied by 0.5, 1.0, and 1.5. Both multipliers are 1.0 in the nominal dynamics.

## C  STATE ABSTRACTION

A $Q^c$-irrelevant abstraction $\sigma\colon S \to S^\dagger$ maps a CMDP to an *abstracted* CMDP (Li et al., 2006). Let $\mathcal{M} = (S, A, P, r, c, d, \gamma)$, and $\omega\colon S \to [0,1]$. The corresponding *abstracted task* is $\mathcal{M}^\dagger = (S^\dagger, A, P^\dagger, r^\dagger, c^\dagger, d, \gamma)$, where

- $P^\dagger(s'^\dagger \mid s^\dagger, a) = \sum_{s \in \sigma^{-1}(s^\dagger)} \sum_{s' \in \sigma^{-1}(s'^\dagger)} \omega(s) P(s' \mid s, a)$,

- $r^\dagger(s^\dagger, a) = \sum_{s \in \sigma^{-1}(s^\dagger)} \omega(s) r(s, a)$,

- $c^\dagger(s^\dagger, a) = \sum_{s \in \sigma^{-1}(s^\dagger)} \omega(s) c(s, a)$,

with $\sigma^{-1}(s^\dagger) = \{s \in S \mid \sigma(s) = s^\dagger\}$ and $\sum_{s \in \sigma^{-1}(s^\dagger)} \omega(s) = 1$, for all $s^\dagger \in S^\dagger$.

## D  CUMULATIVE COSTS IN THE UNCERTAINTY SETS

| $\alpha$ | $\mathcal{M}_1^\diamond$ Worst | 0.1-tail | Avg. | $\mathcal{M}_2^\diamond$ Worst | 0.1-tail | Avg. | $\mathcal{M}_3^\diamond$ Worst | 0.1-tail | Avg. |
|---|---|---|---|---|---|---|---|---|---|
| 0.00 | 805.1 | 780.2 | 316.7 | 862.6 | 856.2 | 268.6 | 536.9 | 457.9 | 89.3 |
| 0.06 | 843.2 | 816.6 | 221.5 | 353.3 | 221.4 | 41.2 | 635.0 | 574.5 | 130.6 |
| 0.12 | 873.5 | 868.2 | 438.1 | 842.8 | 792.9 | 179.2 | **89.7** | **77.2** | **34.2** |
| 0.17 | 780.8 | 683.9 | 150.5 | 692.8 | 399.0 | 110.4 | 725.0 | 569.0 | 95.8 |
| 0.23 | **2.3** | 1.5 | 0.2 | **91.3** | 77.7 | 21.5 | 286.2 | 268.8 | 126.5 |
| 0.29 | 3.2 | **1.1** | **0.1** | 95.7 | **59.7** | **18.1** | 859.0 | 754.0 | 168.2 |
| 0.35 | 134.5 | 100.0 | 11.5 | 271.6 | 162.9 | 58.6 | 833.6 | 809.0 | 190.1 |

Table 2: The worst-case, 0.1-tail, and average expected costs of all the guides trained with *random action noise* within the three source tasks.

| $\alpha$ | $\mathcal{M}_1^\diamond$ Worst | 0.1-tail | Avg. | $\mathcal{M}_2^\diamond$ Worst | 0.1-tail | Avg. | $\mathcal{M}_3^\diamond$ Worst | 0.1-tail | Avg. |
|---|---|---|---|---|---|---|---|---|---|
| 0.0 | 805.1 | 780.2 | 316.7 | 862.6 | 856.2 | 268.6 | 536.9 | 457.9 | 89.3 |
| 0.06 | 880.4 | 871.5 | 537.2 | 769.5 | 717.3 | 108.6 | 219.7 | 210.3 | 49.0 |
| 0.12 | 840.0 | 821.7 | 302.7 | 626.6 | 569.8 | 75.8 | 338.1 | 218.5 | **45.3** |
| 0.17 | 484.0 | 367.2 | 87.7 | 713.9 | 554.1 | 109.7 | **166.4** | **148.3** | 65.5 |
| 0.23 | 790.7 | 774.8 | 372.5 | 773.0 | 678.9 | 97.5 | 459.9 | 335.5 | 102.8 |
| 0.29 | **38.5** | **17.5** | **2.7** | 233.8 | 212.4 | **52.6** | 699.5 | 640.5 | 143.7 |
| 0.35 | 742.5 | 687.9 | 294.2 | **211.4** | **139.0** | 57.2 | 255.3 | 225.7 | 84.7 |

Table 3: The worst-case, 0.1-tail, and average expected costs of all the guides trained with *adversarial perturbations* within the three source tasks.

| $\alpha$ | $\mathcal{M}_1^\diamond$ Worst | 0.1-tail | Avg. | $\mathcal{M}_2^\diamond$ Worst | 0.1-tail | Avg. | $\mathcal{M}_3^\diamond$ Worst | 0.1-tail | Avg. |
|---|---|---|---|---|---|---|---|---|---|
| 0.0 | **147.4** | **82.9** | **9.7** | 668.7 | 563.6 | 108.2 | 892.7 | 881.7 | 262.4 |
| $10^{-3}$ | 594.2 | 139.0 | 13.2 | **551.1** | **423.1** | **59.3** | **877.6** | **871.0** | **186.4** |

Table 4: The worst-case, 0.1-tail, and average expected costs of all the guides trained with *entropy maximization* within the three source tasks.

## E  SAFETY JUMP-START TABLES

| $\alpha$ | $\mathcal{M}_1^\diamond$ Worst | 0.1-tail | Avg. | $\mathcal{M}_2^\diamond$ Worst | 0.1-tail | Avg. | $\mathcal{M}_3^\diamond$ Worst | 0.1-tail | Avg. |
|---|---|---|---|---|---|---|---|---|---|
| 0.06 | -60.4 | -62.1 | -86.6 | **-56.3** | **-63.0** | **-87.4** | **-48.9** | **-55.1** | **-82.9** |
| 0.12 | -49.6 | -54.3 | -80.9 | -31.4 | -39.2 | -84.1 | -41.5 | -53.6 | -80.2 |
| 0.17 | -64.3 | -69.8 | -87.9 | -32.4 | -51.2 | -83.9 | -36.6 | -47.6 | -80.8 |
| 0.23 | **-71.2** | **-80.3** | **-91.9** | -34.8 | -49.7 | -85.2 | -23.7 | -39.1 | -75.0 |
| 0.29 | -62.8 | -65.2 | -86.5 | -55.8 | -59.5 | -87.3 | -15.6 | -25.7 | -76.4 |
| 0.35 | -49.3 | -61.8 | -87.5 | -5.8 | -35.7 | -79.6 | -29.7 | -38.6 | -78.9 |

Table 5: *Deterministic students* with the *random action noise* guide.

| $\alpha$ | $\mathcal{M}_1^\diamond$ Worst | 0.1-tail | Avg. | $\mathcal{M}_2^\diamond$ Worst | 0.1-tail | Avg. | $\mathcal{M}_3^\diamond$ Worst | 0.1-tail | Avg. |
|---|---|---|---|---|---|---|---|---|---|
| 0.06 | 179.7 | 134.9 | -17.6 | **50.6** | **13.1** | -63.0 | **18.8** | **2.7** | **-57.9** |
| 0.12 | 263.5 | 183.3 | 15.9 | 138.2 | 105.3 | -47.3 | 64.5 | 40.4 | -50.6 |
| 0.17 | 156.9 | 101.6 | -20.3 | 123.1 | 53.2 | -51.3 | 70.1 | 29.1 | -53.1 |
| 0.23 | **46.7** | **18.7** | **-52.3** | 117.4 | 47.8 | -56.1 | 68.5 | 48.0 | -47.0 |
| 0.29 | 195.0 | 126.4 | -18.8 | 86.2 | 26.8 | **-63.7** | 126.7 | 105.7 | -41.1 |
| 0.35 | 205.1 | 103.8 | -28.7 | 135.2 | 92.8 | -41.1 | 116.2 | 63.2 | -51.6 |

Table 6: *Nondeterministic students* with the *random action noise* guide.

| $\alpha$ | $\mathcal{M}_1^\diamond$ Worst | 0.1-tail | Avg. | $\mathcal{M}_2^\diamond$ Worst | 0.1-tail | Avg. | $\mathcal{M}_3^\diamond$ Worst | 0.1-tail | Avg. |
|---|---|---|---|---|---|---|---|---|---|
| 0.06 | -50.8 | -57.0 | -82.7 | -44.9 | -52.4 | -84.1 | -41.7 | -49.0 | -79.8 |
| 0.12 | -56.6 | -63.0 | -87.2 | -31.3 | -39.9 | -83.2 | -39.7 | -51.2 | -81.3 |
| 0.17 | **-68.4** | **-72.1** | **-88.6** | 17.3 | -28.9 | -78.8 | -41.8 | -52.4 | -80.8 |
| 0.23 | -52.7 | -64.4 | -85.7 | -19.9 | -29.9 | -80.5 | -49.7 | -58.6 | -83.2 |
| 0.29 | -62.0 | -69.4 | -87.8 | -31.8 | -49.4 | -83.5 | **-52.3** | **-59.1** | **-82.9** |
| 0.35 | -65.0 | -67.6 | -80.5 | **-61.0** | **-68.2** | **-88.4** | -48.8 | -52.4 | -79.9 |

Table 7: *Deterministic students* with the *adversarially trained* guide.

| $\alpha$ | $\mathcal{M}_1^\diamond$ Worst | 0.1-tail | Avg. | $\mathcal{M}_2^\diamond$ Worst | 0.1-tail | Avg. | $\mathcal{M}_3^\diamond$ Worst | 0.1-tail | Avg. |
|---|---|---|---|---|---|---|---|---|---|
| 0.06 | 177.7 | 143.1 | 0.2 | 151.5 | 86.9 | -50.1 | 61.7 | 30.8 | -52.5 |
| 0.12 | 106.3 | 88.9 | -30.8 | 142.1 | 102.6 | -46.3 | 43.0 | 25.6 | -54.8 |
| 0.17 | **73.6** | **53.4** | **-36.2** | 161.0 | 122.9 | -36.3 | **21.2** | 13.4 | -54.3 |
| 0.23 | 197.2 | 125.7 | -14.2 | 135.2 | 99.9 | -44.9 | 40.9 | 21.3 | -55.6 |
| 0.29 | 188.7 | 101.0 | -23.7 | 86.6 | 36.1 | -55.4 | 24.1 | **1.3** | **-59.4** |
| 0.35 | 172.2 | 112.2 | 10.1 | **56.3** | **26.5** | **-63.5** | 74.1 | 39.4 | -50.2 |

Table 8: *Nondeterministic students* with the *adversarially trained* guide.

| $\alpha$ | $\mathcal{M}_1^\diamond$ Worst | 0.1-tail | Avg. | $\mathcal{M}_2^\diamond$ Worst | 0.1-tail | Avg. | $\mathcal{M}_3^\diamond$ Worst | 0.1-tail | Avg. |
|---|---|---|---|---|---|---|---|---|---|
| 0.0 | -69.3 | -75.5 | -90.8 | -34.3 | -58.5 | -86.2 | 12.3 | -5.0 | -71.8 |
| 0.001 | -44.0 | -66.2 | -89.4 | -30.1 | -54.9 | -84.8 | -0.7 | -20.5 | -73.6 |

Table 9: *Deterministic students* with the *entropy maximizing* guide.

| $\alpha$ | $\mathcal{M}_1^\diamond$ | | | $\mathcal{M}_2^\diamond$ | | | $\mathcal{M}_3^\diamond$ | | |
|---|---|---|---|---|---|---|---|---|---|
| | Worst | 0.1-tail | Avg. | Worst | 0.1-tail | Avg. | Worst | 0.1-tail | Avg. |
| $\alpha$ | Worst | 0.1-tail | Avg. | Worst | 0.1-tail | Avg. | Worst | 0.1-tail | Avg. |
| 0.0 | 44.9 | 29.1 | -46.3 | 80.3 | 45.3 | -57.2 | 132.1 | 91.5 | -40.1 |
| 0.001 | 165.2 | 77.6 | -37.4 | 101.6 | 73.8 | -51.4 | 137.8 | 108.3 | -36.0 |

Table 10: *Nondeterministic students* with the *entropy maximizing* guide.

# F ROBUSTNESS HEATMAPS FROM THE SOURCE TASK

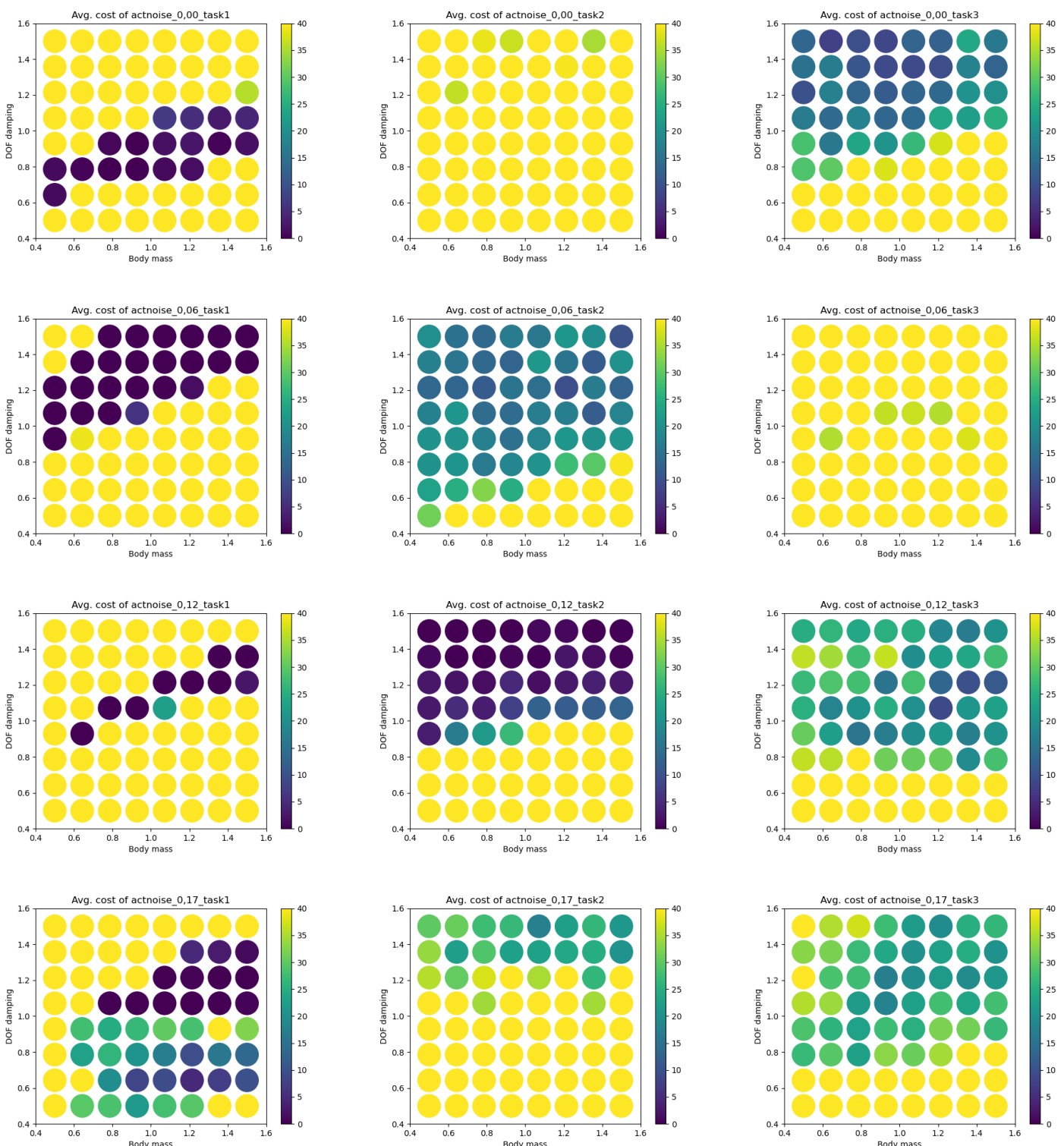

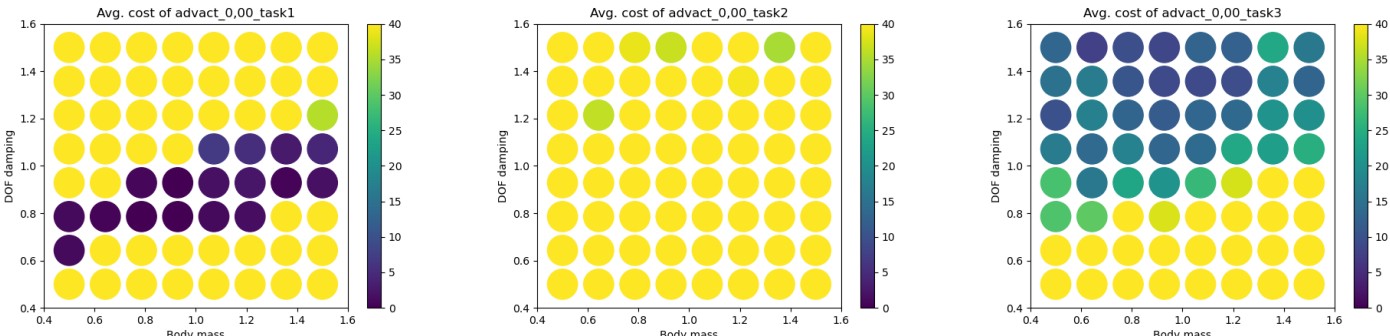

Figure 12: Average expected costs of all the guides trained with *random action noise* within the three source tasks, where each dot represents a different dynamics function.

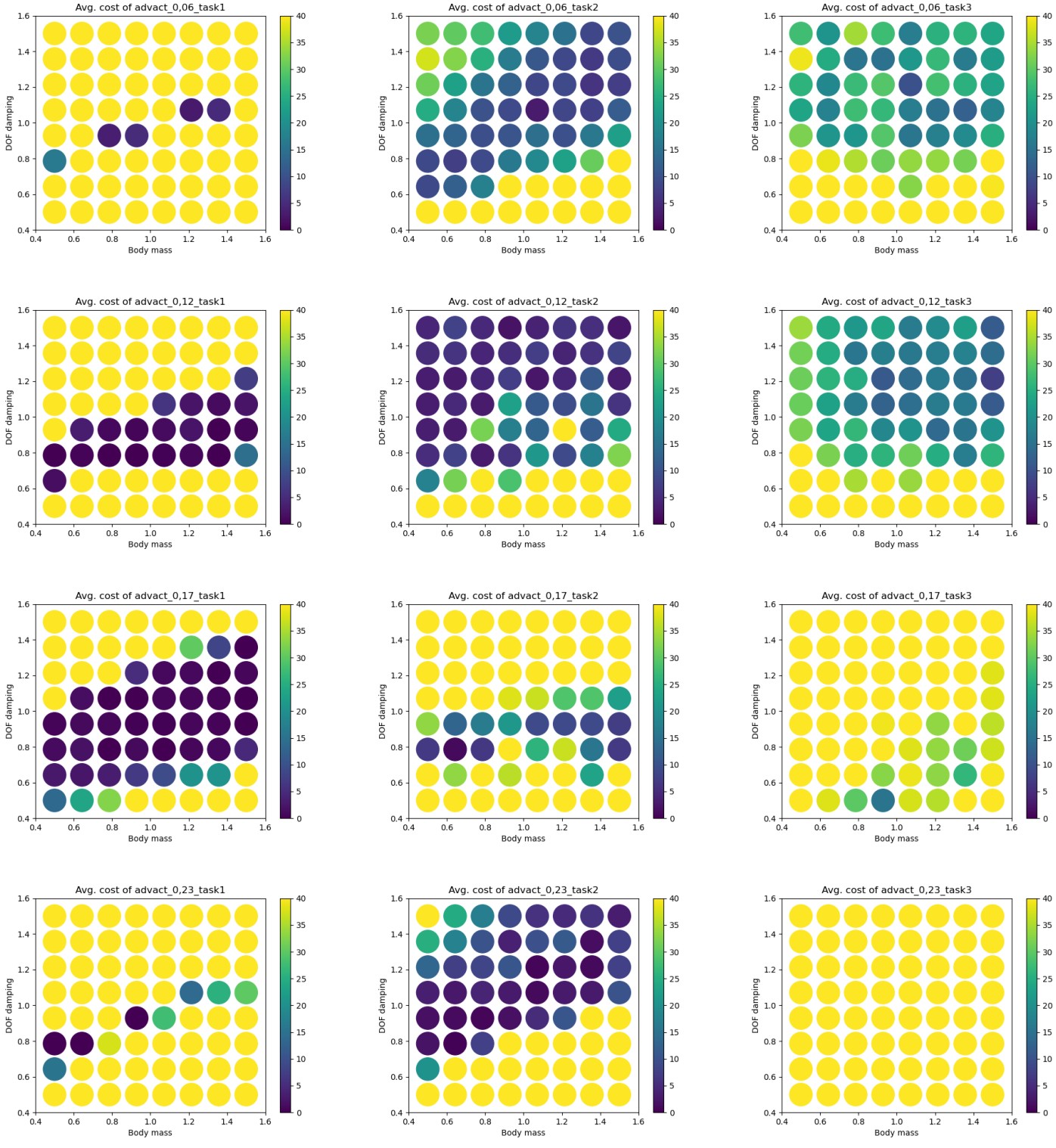

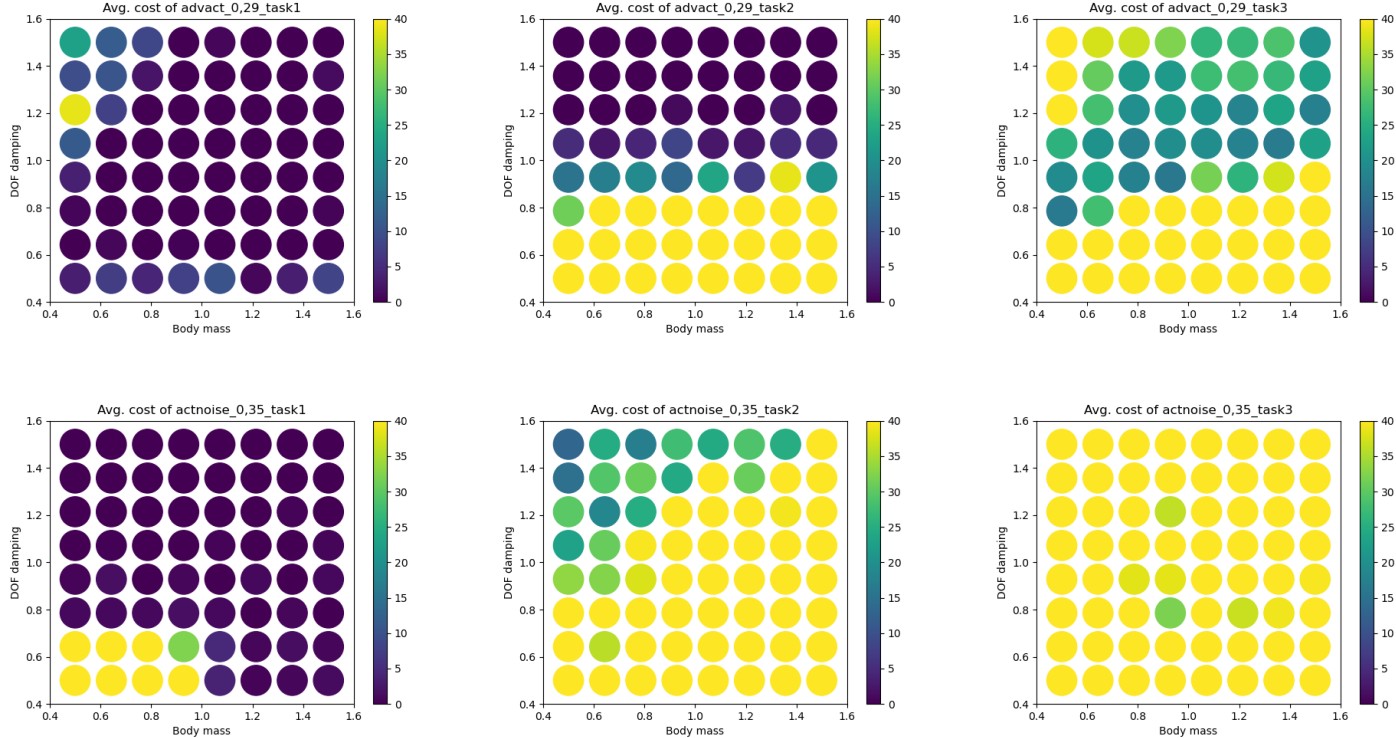

Figure 13: Average expected costs of all the guides trained with *adversarial perturbations* within the three source tasks, where each dot represents a different dynamics function.

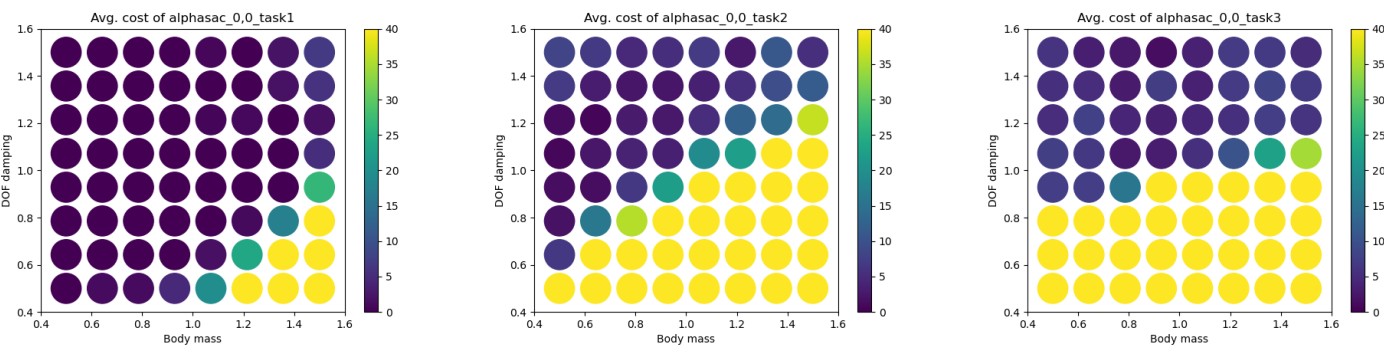

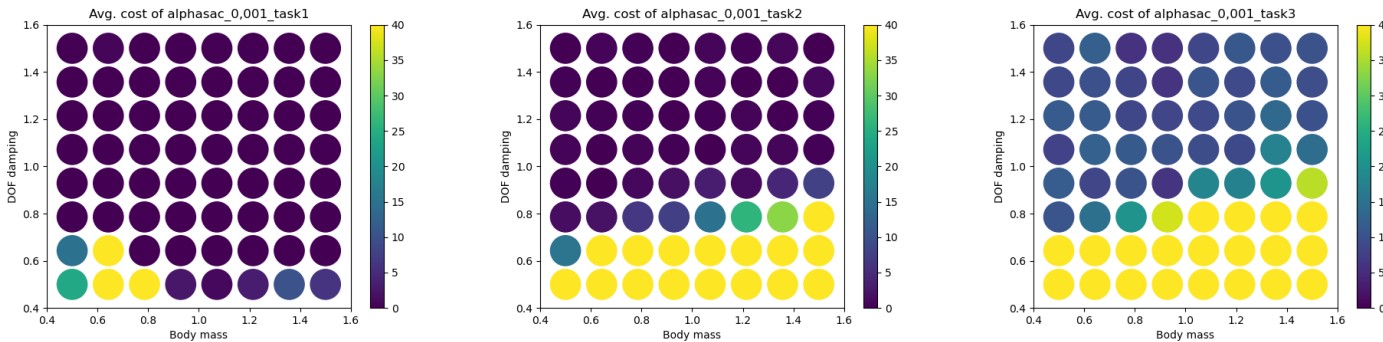

Figure 14: Average expected costs of all the guides trained with *entropy maximization* within the three source tasks, where each dot represents a different dynamics function.

## G    SAFETY JUMP-START HEATMAPS

### G.1    TRANSFERRING TO THE NONDETERMINISTIC STUDENT

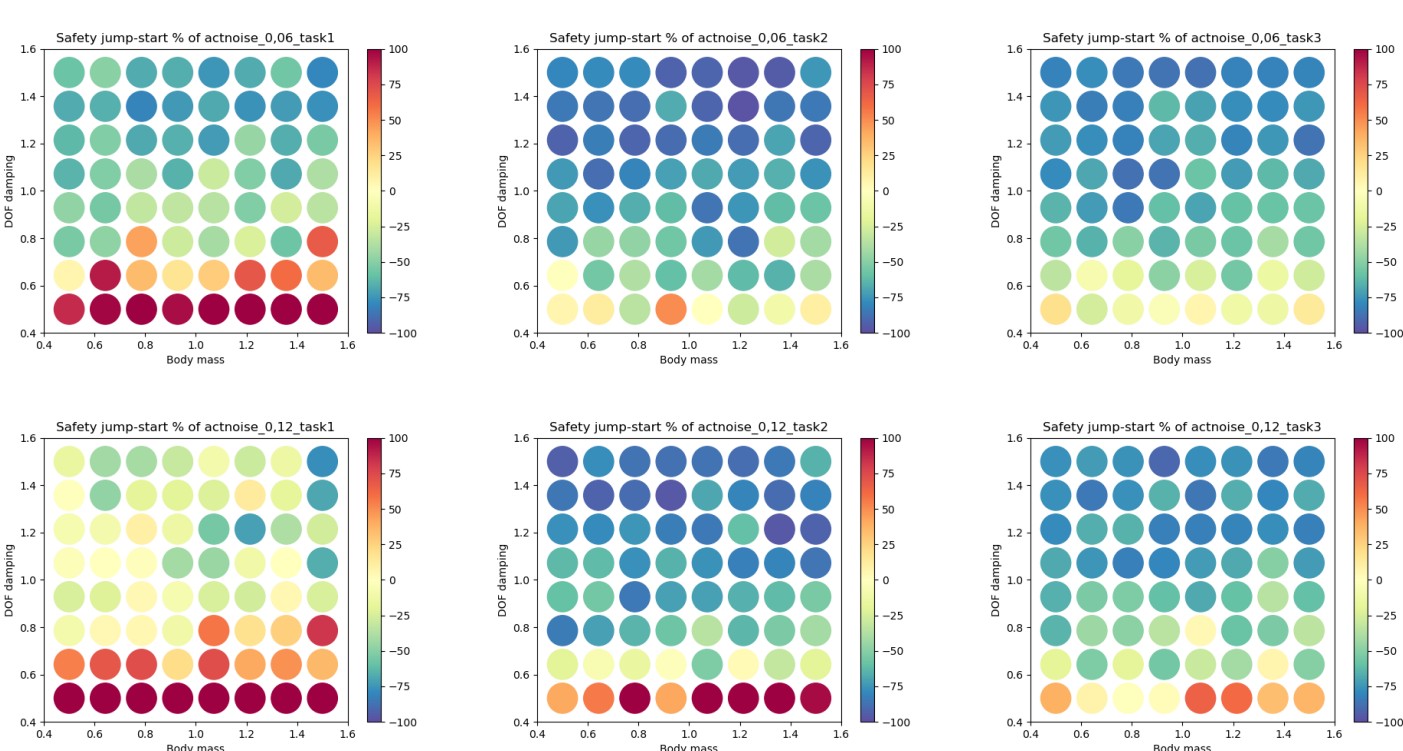

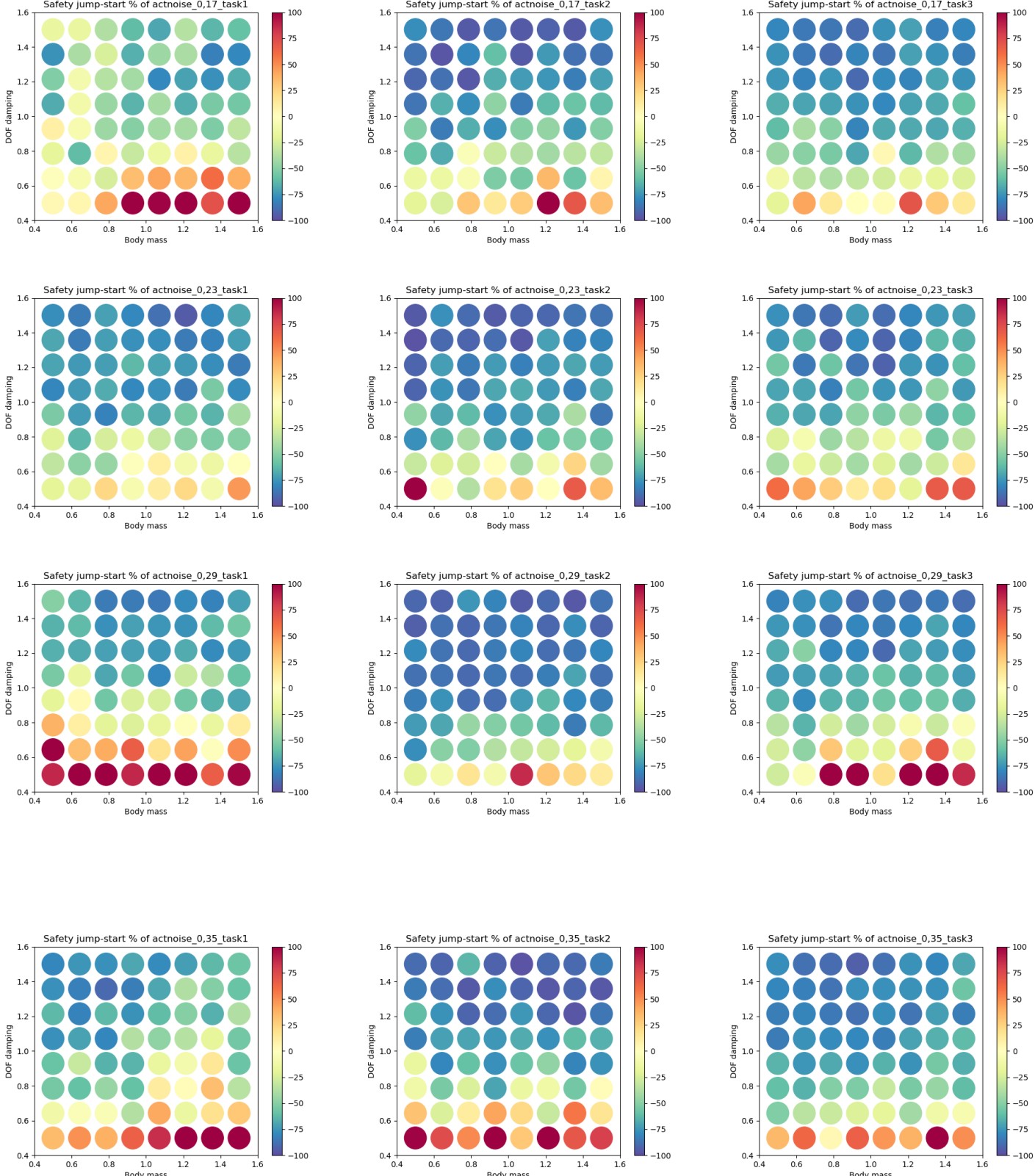

Figure 15: Safety jump-starts of the nondeterministic students with the guides trained with *random action noise* within the three target tasks, where each dot represents a different dynamics function.

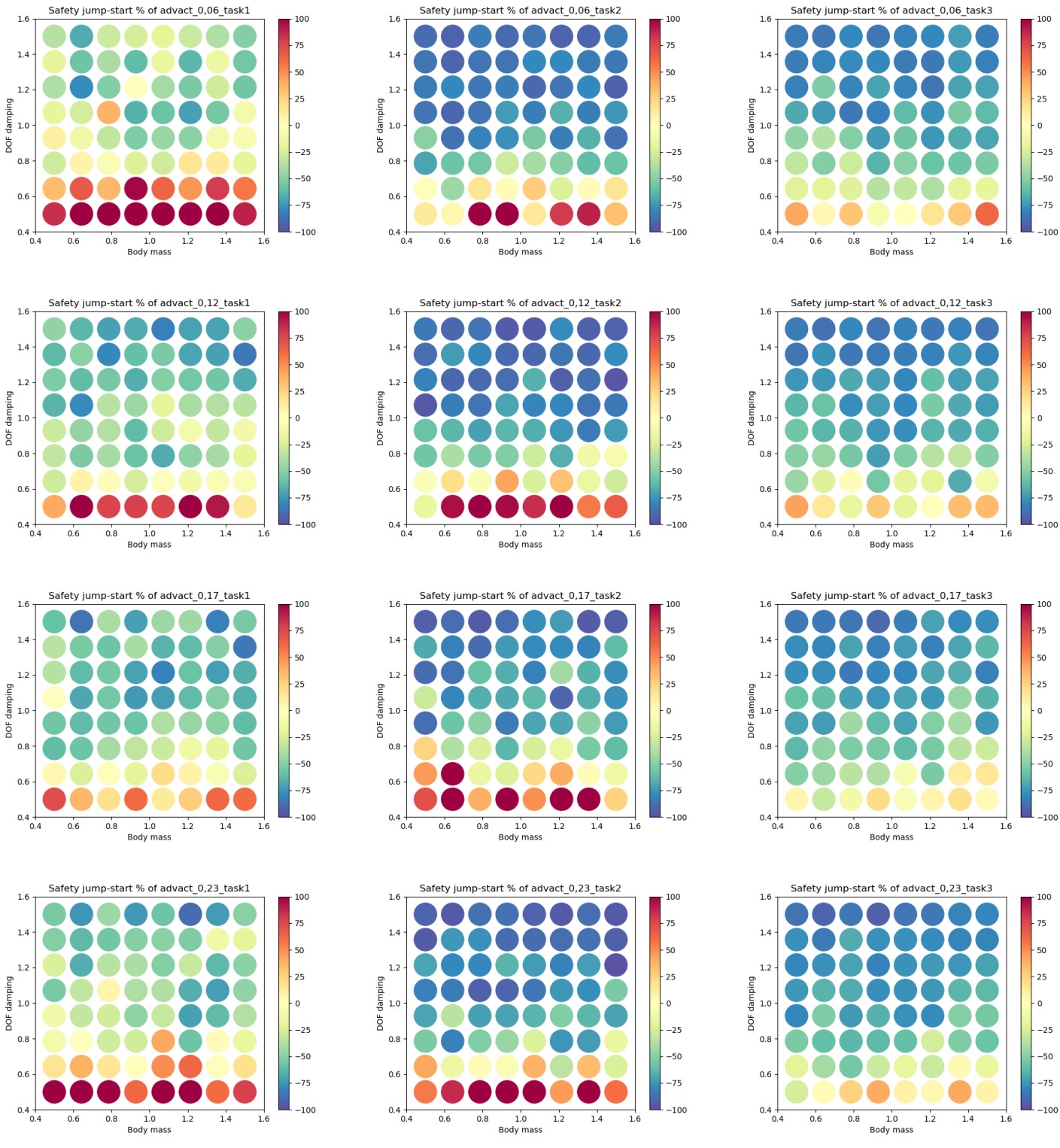

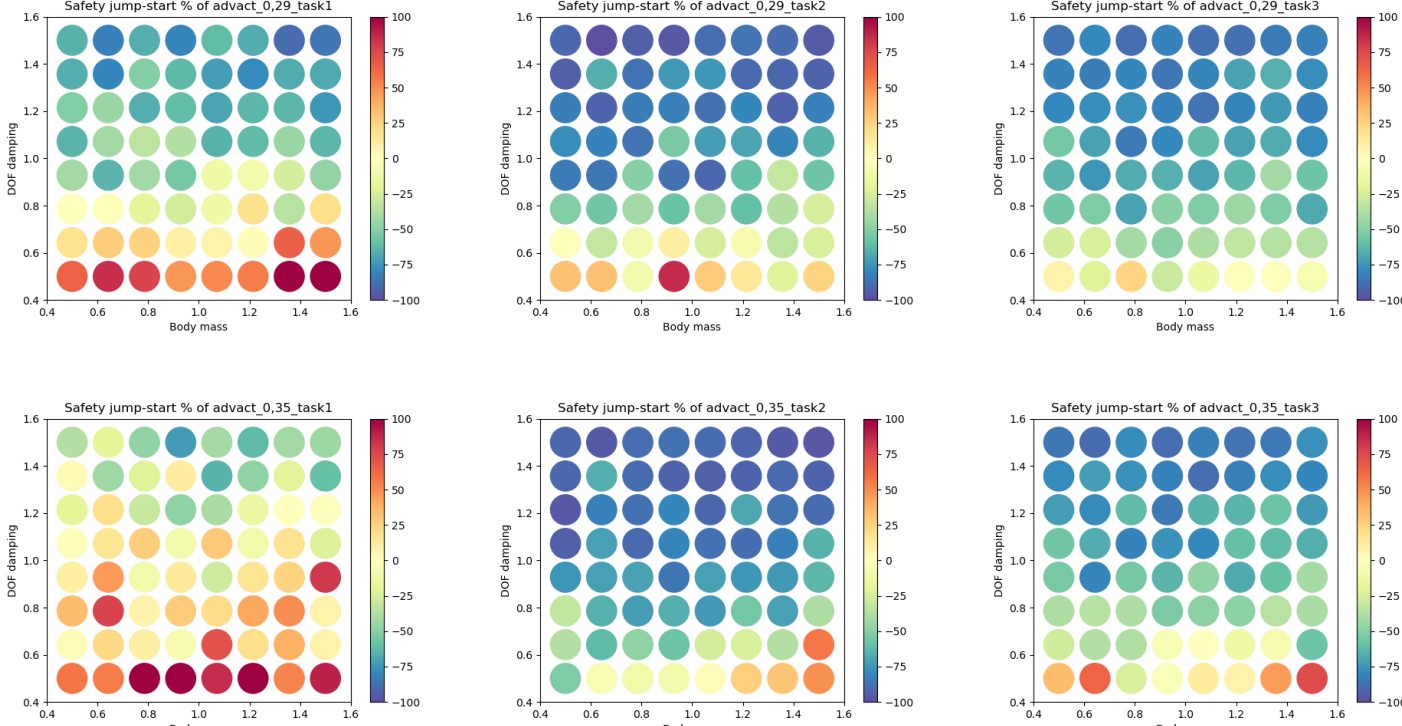

Figure 16: Safety jump-starts of the nondeterministic students with the guides trained with *adversarial perturbations* within the three target tasks, where each dot represents a different dynamics function.

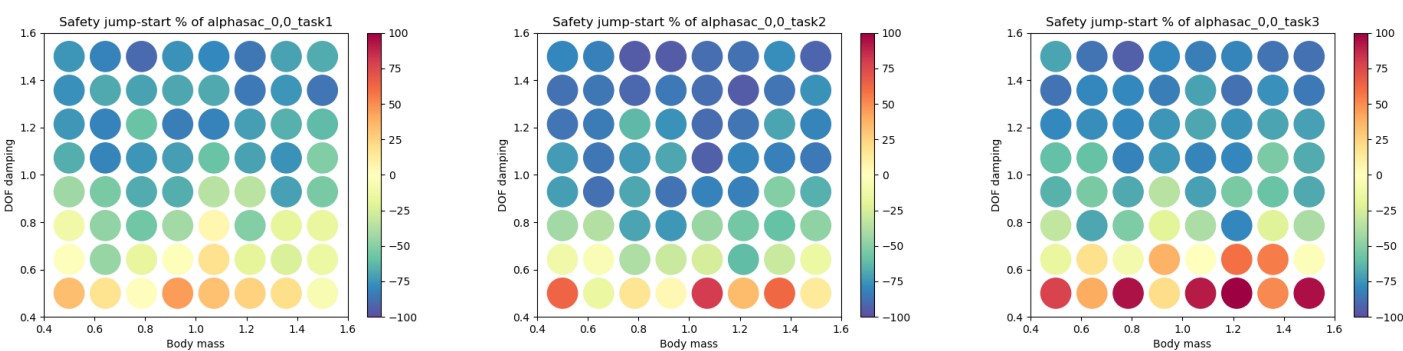

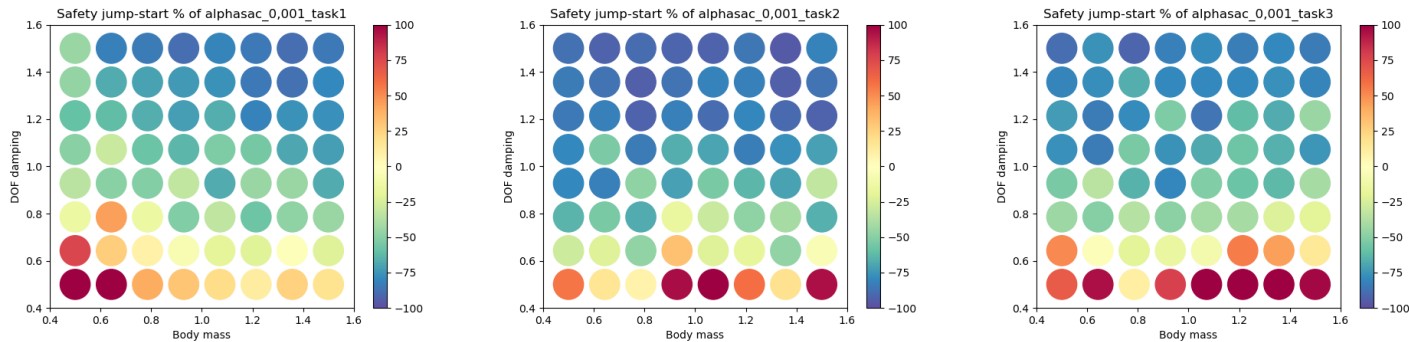

Figure 17: Safety jump-starts of the nondeterministic students with the guides trained with *entropy maximization* within the three target tasks, where each dot represents a different dynamics function.

## G.2 TRANSFERRING TO THE DETERMINISTIC STUDENT

Figure 18: Safety jump-starts of the deterministic students with the guides trained with *random action noise* within the three target tasks, where each dot represents a different dynamics function.

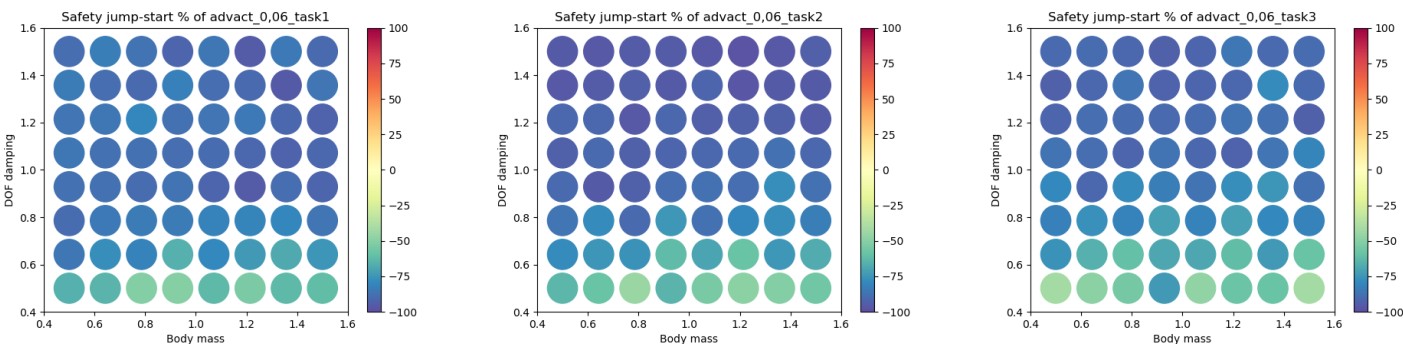

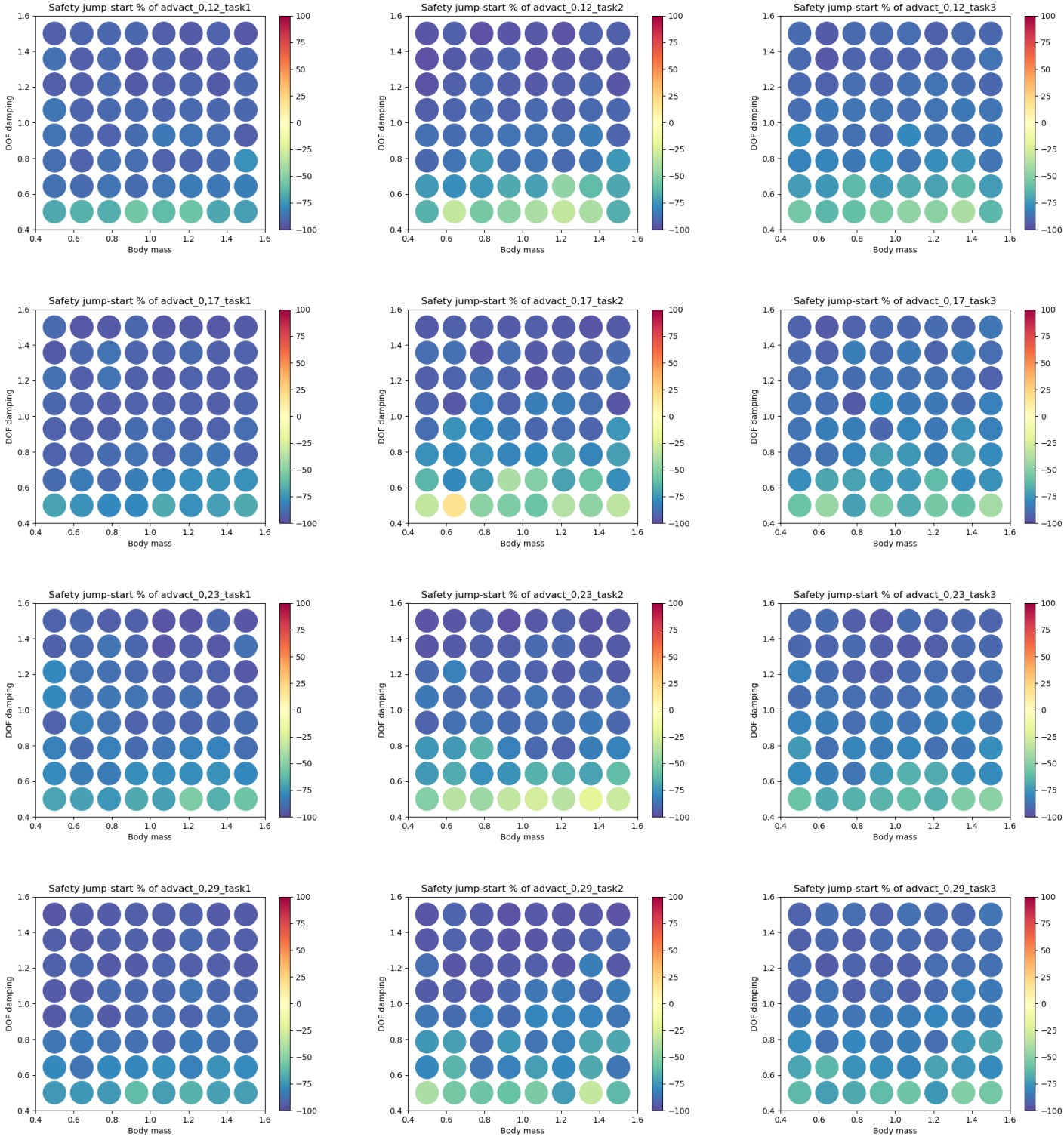

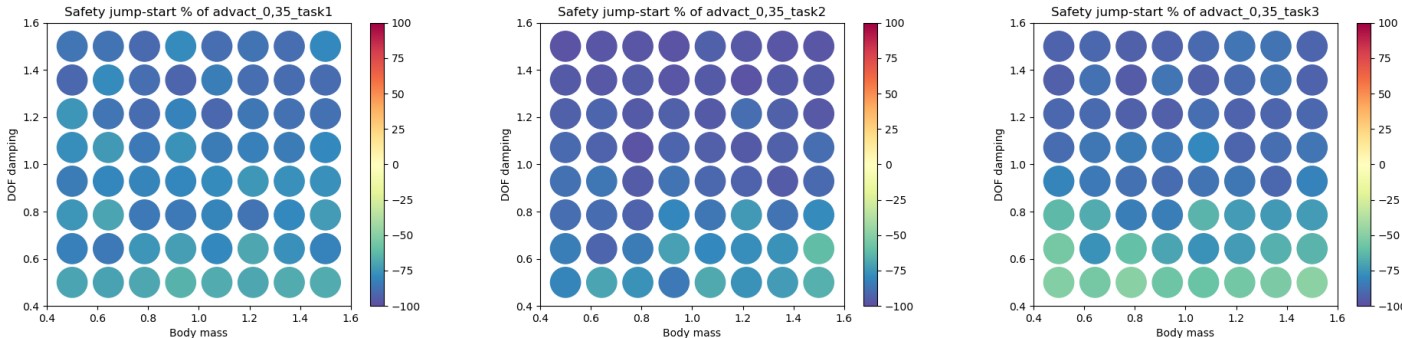

Figure 19: Safety jump-starts of the deterministic students with the guides trained with *adversarial perturbations* within the three target tasks, where each dot represents a different dynamics function.

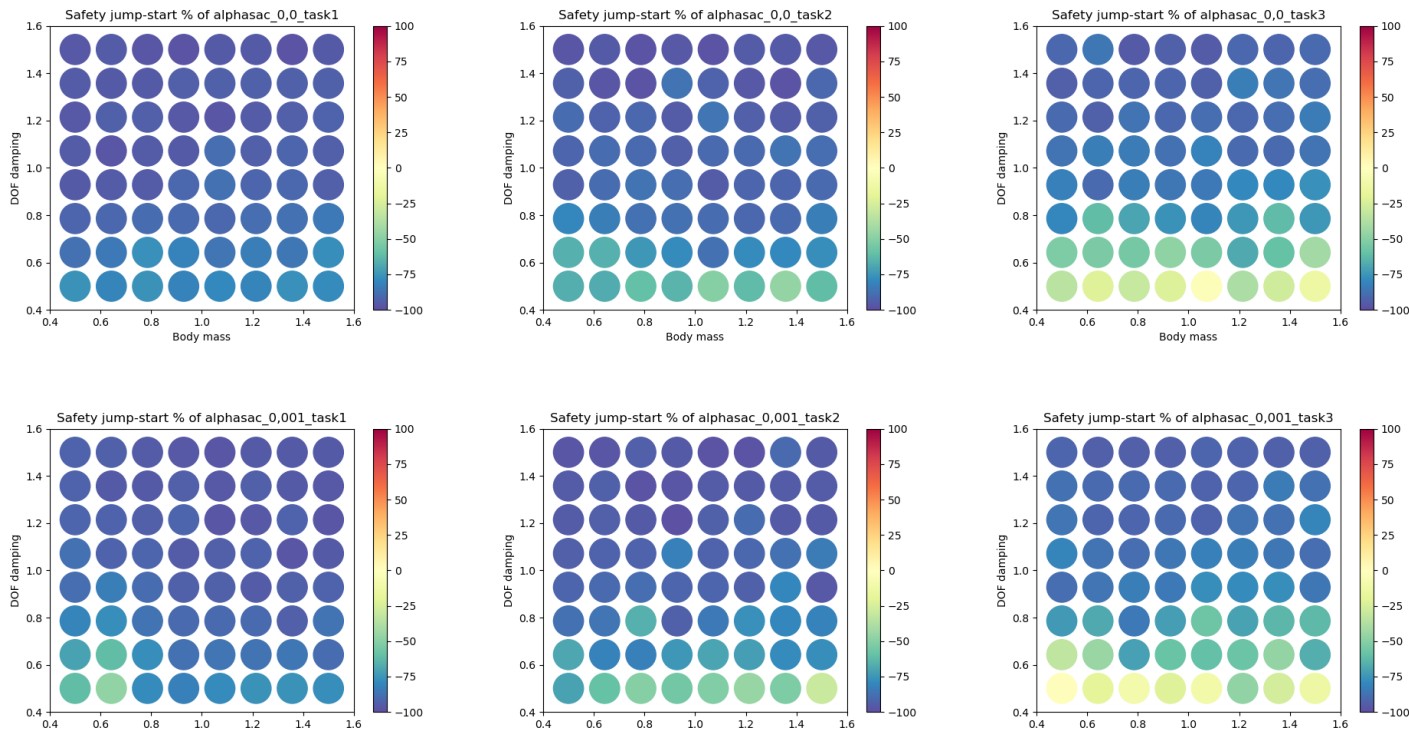

Figure 20: Safety jump-starts of the deterministic students with the guides trained with *entropy maximization* within the three target tasks, where each dot represents a different dynamics function.

