# OpenReview forum: "Robust Transfer of Safety-Constrained Reinforcement Learning Agents"
_ICLR.cc/2025/Conference — ICLR 2025 Poster_

### Official Review · Reviewer_LcYf · 2024-10-17

**Soundness:** 3
**Presentation:** 2
**Contribution:** 2
**Rating:** 6
**Confidence:** 3

**Summary:**

This paper addresses a problem where a policy is trained in a controlled environment and then transferred  to the real environment. While safety violation is allowed in a controlled environment, unsafe actions may be catastrophic in the real environment. Unlike the previous work that allow policy transfers as long as the new environment preserves safety-related dynamics, this paper explicitly incorporate the dynamics differences or shifts. The authors propose a method to robustifies an agent in the controlled environment and then provably achieves a safe transfer to new environments. The empirical evaluation shows the effectiveness of their proposed method.

**Strengths:**

- The authors address a very important and interesting problem. I think such problems are fit to some real-world safety-critical problems.
- The Related Work section is concise but well-written and easy to follow.
- The Background section is also easy to follow and well-presented.
- Assumptions are all reasonable and indeed mild.
- I think Theorem 1 is good. The proof seems to be correct and easy to follow.
- Definitions on safety jump-start, $\Delta$ time to safety, and a safe transfer-learning metric are well-presented despite their complicated notions.
- The resulting algorithm presented in Section 5 is technically sound and easy to understand.

**Weaknesses:**

- In Related work section, I think there is a missing citations:
    - Zhang, Jesse, et al. "Cautious adaptation for reinforcement learning in safety-critical settings." International Conference on Machine Learning. PMLR, 2020.
- Experimental results seem to include only ablation study. I am wondering why Yang et al. (2023) is not compared with their proposed method. If it has been already included as a baseline method, please clarify in the experiment section.
- I think the experimental settings have not fully written in the main paper and appendix. Given the source code is not attached, I feel the reproducibility of this paper is low.

**Questions:**

- Q1: In Definition 2, what is the definition of $\omega$? I know that the authors mention that the sum of $\omega$ is equal to 1, but I think it not uniquely defined.
- Q2: Is it possible to compare the authors' proposed method with Yang et al. (2023)? As the authors mention, this previous work seems to be the state-of-the-art method if I understand correctly.

Since I think this paper is overall of high quality, I am happy to increase the score from the current one (i.e., 5: borderline reject) when my concerns are addressed during the rebuttal period.

---

> ### Author Response · Authors · 2024-11-23
> **Response to Reviewer LcYf**
>
> We would like thank reviewer LcYf  for the insightful feedback, which has greatly helped us improve the paper.
>
> - **(Q)uestion1.** _What is the definition of $\omega$?_
>
> $\omega$ is a probability measure on the Borel sets of $S$, referred to as the "weighting function", which we need for a well-defined abstracted task (Li et al., 2006).
> In practice, this is usually a uniform distribution over the state-space, but indeed it is not uniquely defined and can thus have other shapes.
>
> We will include a discussion in the updated version of the paper.
>
> > - Lihong Li, Thomas J. Walsh, and Michael L. Littman. Towards a unified theory of state abstraction for MDPs. In AI&M, 2006.
>
>
> - **(W)eakness2 and Q2.** _Is it possible to compare the proposed method with Yang et al. (2023)?_
>
> Yes, Yang et al. (2023) is the state-of-the-art, which yields non-robust guides.
> The comparison between non-robust guides and our (robust) guides is shown in Fig 7.
> Our paper would benefit from making this clear.
>
> - **W1.** _In Related work section, I think there is a missing citations:
> Zhang, Jesse, et al. "Cautious adaptation for reinforcement learning in safety-critical settings." International Conference on Machine Learning. PMLR, 2020._
>
> We thank the reviewer for this reference and will of course include the citation together with an appropriate discussion.
>
>
> - **W3.** _Given the source code is not attached, reproducibility is low._
>
> Please refer to our global response **General Comment 5**.
>
> We hope this clarifies the reviewer's questions. We are happy to follow up if something is not clear.

---

> > ### Comment · Reviewer_LcYf · 2024-11-23
> > **Thank you for clarification**
> >
> > Thank you for clarifications! I think the authors are likely to address my concerns at the time of initial review. I would like to finalize my recommendation after seeing the revised version. Could you upload it during this discussion phase?

---

> > > ### Comment · Reviewer_LcYf · 2024-11-27
> > >
> > > Thank you for uploading a revised version.
> > > Could you tell me how the authors revised this paper regarding this point?
> > >
> > > > Yes, Yang et al. (2023) is the state-of-the-art, which yields non-robust guides. The comparison between non-robust guides and our (robust) guides is shown in Fig 7. Our paper would benefit from making this clear.

---

> > > > ### Author Response · Authors · 2024-11-27
> > > > **Response to Reviewer LcYf**
> > > >
> > > > Thank you for following up on the discussion. We have uploaded a revised version of the paper, updating the caption of Fig. 7 and clarifying its reference in line 521. In short, we already compare our algorithm with the method proposed by Yang et al (2023) and now we made this more explicit in the paper. Please let us know if you have any further questions.

---

> ### Comment · Reviewer_LcYf · 2024-11-28
>
> I consider that the current version presents empirical results much more clearly than the initial version. I changed the score accordingly and now slightly lean on acceptance.

---

### Official Review · Reviewer_Vwu8 · 2024-10-29

**Soundness:** 3
**Presentation:** 3
**Contribution:** 4
**Rating:** 6
**Confidence:** 4

**Summary:**

This paper addresses a critical challenge in safe reinforcement learning: how to transfer an agent trained in a controlled environment to a real-world setting with different dynamics while maintaining safety guarantees. It propose a methodology that makes the transfer process robust against variations in dynamics between source and target environments.

**Strengths:**

The theoretical work is sound and well-constructed. Theorem 1, which establishes safety guarantees under transfer, is particularly well-developed and properly scoped. The proofs are thorough without being needlessly complex, though some of the assumptions (like the existence of Q-irrelevant abstractions) might prove limiting in practice. The authors demonstrate that their robustified agents can achieve substantial improvements in worst-case performance, with some configurations reaching complete safety transfer. The empirical validation is thorough and well-designed. The authors test their approach across multiple environments and provide meaningful comparisons between different robustification techniques. The ablation studies effectively isolate the contributions of individual components, though more complex environments could have strengthened the conclusions further. Overall, the paper makes a worthwhile contribution to the field of safe reinforcement learning.

**Weaknesses:**

The paper exhibits some theoretical limitations in its treatment of safety-preserving abstractions and uncertainty quantification. The foundational assumption of Q-irrelevant abstractions $\sigma: S_\odot \rightarrow S_\diamond$ being continuous and bounded (Assumption 2) might become problematic for hybrid or discontinuous systems, while the bounded uncertainty set assumption $\exists\varepsilon \in \mathbb{R}, \forall P, P' \in U_\odot, \|P - P'\| \leq \varepsilon$ fails to capture fat-tailed uncertainty distributions common in real systems. The robustification approach through entropy maximization $r^\diamond_t(s_t, a_t) = r^b_t(s_t, a_t) + \alpha r^H_t(s_t, a_t)$ lacks guarantees about value function Lipschitz constants, potentially compromising stability under state-space perturbations. Moreover, the control switching mechanism $\pi_b(s_t) = \begin{cases} (\pi^\diamond \circ \sigma)(s_t) & \text{if } c_{t-1} > 0 \\ \pi_\odot(s_t) & \text{else} \end{cases}$ introduces potential chattering issues near constraint boundaries, while the p-tail metric $M_{\leq p}(M_\odot, \pi) = \int_B M(M^\odot_P, \pi) dP$ requires additional measure-theoretic assumptions that remain unjustified.

The empirical validation methodology demonstrates concerning limitations in both scope and rigor. The discretization of the uncertainty set using only $N=8$ values for mass and friction parameters provides insufficient coverage of potential failure modes, while the safety jump-start metric $J(\pi^\diamond, M_\odot) = \mathbb{E}[\frac{c(\tau) - c(\tau')}{c(\tau')} | \tau \sim \rho_1, \tau' \sim \rho'_1]$ masks potentially dangerous temporal patterns through trajectory averaging. The environments tested $(M^\diamond_1, M^\diamond_2, M^\diamond_3)$ share similar underlying physics, failing to challenge the framework with fundamentally different dynamics structures. The paper's treatment of model parameterization through simple mass and friction variations $m_i = (0.5 + \frac{i-1}{7})m^\diamond$ and $\eta_i = (0.5 + \frac{i-1}{7})\eta^\diamond$ inadequately captures the complexity of real-world dynamic variations, particularly in systems with coupled parameters or higher-order effects. Furthermore, the Lagrangian multiplier approach for policy imitation lacks theoretical convergence guarantees when combined with robustification techniques, potentially leading to unstable learning dynamics.

**Questions:**

- In Section 4.1, you assume the existence of a Q-irrelevant abstraction σ that is continuous and bounded. Could you elaborate on how this abstraction can be constructed in practice for high-dimensional state spaces? A concrete example or algorithm would significantly strengthen the paper's practical applicability.

- The bounded uncertainty set assumption (Assumption 3) seems crucial for your theoretical guarantees. Could you provide insights on how to estimate the bound ε in practice?

- Your empirical evaluation uses N=8 values for discretizing the uncertainty set. Could you justify this choice of granularity? And discuss how discretization errors affect your worst-case guarantees?

- The control switching mechanism π_b(s_t) might lead to chattering. Have you analyzed the frequency of policy switches during transfer?

---

> ### Author Response · Authors · 2024-11-23
> **Response to Reviewer Vwu8 (1)**
>
> We sincerely thank reviewer Vwu8 for the thoughtful and constructive feedback.
> Your insights into both the theoretical aspects and the empirical validation of our work have significantly helped us improve the quality of the paper.
>
> - **(W)eakness1.** _$\sigma$ being continuous and bounded might become problematic for discontinuous and hybrid systems._
>
> That is indeed correct.
> We will omit the mention of continuity in the revised version,
> as it is sufficient for $\sigma$ to preserve boundedness for the theorem to hold.
> Moreover, the theorem remains valid even for discrete systems, provided the state space is a metric space.
> However, our focus is on continuous systems, as our method is specifically designed with continuous dynamics in mind.
>
>
> - **(Q)uestion1.** _About the construction of $\sigma$ in practice for high-dimensional state spaces._
>
> Please refer to our global response **General Comment 1** for clarifications about this point.
>
> - **Q2.** _How is the bound $\varepsilon$ estimated in practice?_
>
> Please refer to our global response **General Comment 3** for clarifications about this point.
>
> - **Q3 and W6.** _The discretization of the uncertainty set using only $N = 8$ values may be insufficient. Could you justify this choice?_
>
> To determine $N$, we carried out a quantitative and qualitative analysis on the relationship between the estimated worst-case dynamics and $N$, and found that $N=8$ is sufficient.
> This is reasonable as our uncertainty sets are relatively simple. Settings with more complex uncertainty sets and cost functions would require finer discretization.
>
> - **W2.** _The bounded uncertainty set assumption $\exists \varepsilon \in \mathbb{R}, \forall P, P' \in U^\odot, |P - P'| \leq \varepsilon$ fails to capture fat-tailed uncertainty distributions common in real systems._
>
> The restriction that $U$ is bounded indeed disallows fat-tailed uncertainty sets. However, we would like to highlight that
> 1. Similar assumptions are relatively common in this kind of setting. For example, Eysenbach and Levine (2022) show that robustness achieved through entropy maximization results in a policy that is safe within a ball of radius $\epsilon > 0$ around the nominal dynamics.
> 2. Our method can still be used to obtain robust policies in settings where $U$ is unbounded, even if some safety guarantees do not apply.
> 3. It is challenging to provide safety guarantees without any assumptions on the shape of $U$, as an unbounded $U$ would allow tasks that are infinitely different from each other.
>
> > - Eysenbach, B. and Levine, S. (2022). Maximum entropy RL (provably) solves some robust RL problems. In ICLR. OpenReview.net.
>
>
> - **W3.** _The robustification approach through entropy maximization $r^\diamond(s, a) = r^b(s, a) + \alpha r^H(s, a)$ lacks guarantees about value function Lipschitz constants, potentially compromising stability under state-space perturbations._
>
> We would like to ask the reviewer to clarify what they mean by stability under state-space perturbations and how that relates to our approach.
>
> Our understanding is that entropy maximization indeed lacks guarantees regarding the Lipschitz continuity of the value function. However, this is an issue with the technique of entropy maximization itself, while 1. entropy maximization is a well-established technique in the area of robust RL (Moos et al. 2022) and 2. our method does not solely consist of entropy maximization, as this is just one of the three robustification techniques that we examined.
>
> > - Moos, J., Hansel, K., Abdulsamad, H., Stark, S., Clever, D., and Peters, J. (2022). Robust reinforcement learning: A review of foundations and recent advances. Mach. Learn. Knowl. Extr., 4(1):276–315.
>
> - **W4.** _The control switching mechanism $\pi(s) = (\pi^\diamond \circ \sigma)(s)$ if $c > 0$ else $\pi^\odot(s)$ introduces potential chattering issues near constraint boundaries_
>
> The control switch approach changes control from the student to the guide at most once per episode.
> We will make this clear in the paper.
>
> - **W5.** _The p-tail metric $M^{\leq p}(M^\odot, \pi) = \int_B M(M^\odot_P, \pi) dP$ requires additional measure-theoretic assumptions that remain unjustified._
>
> The current definition in the paper is somewhat clunky, and we will update it in the next version.
> Nevertheless, the $p$-tail does not introduce new assumptions.
>
> Let's fix $p = 0.1$ for simplicity. Then, this
> definition simply describes the mean of the top 10\%
> highest costs.
> In theory, it's the integral as defined in the paper, while in practice, it is a simple statistic.

---

> > ### Comment · Reviewer_Vwu8 · 2024-11-25
> > **Thank you for clarification**
> >
> > Thank you for addressing my comments. The authors have provided clear responses, and my concerns have been resolved.

---

> ### Author Response · Authors · 2024-11-23
> **Response to Reviewer Vwu8 (2)**
>
> - **W7.** _The safety jump-start metric $J(\pi^\diamond, M^\odot) = \mathbb{E} [\frac{c(\tau) - c(\tau')}{c(\tau')}]$ masks potentially dangerous temporal patterns through trajectory averaging._
>
> Thanks for bringing this up.
> We would like to point out that the notion of safety in the constrained RL already suffers with this issue.
> To mitigate such situations we would need to consider the worst-case trajectories, for instance by taking into account the CVaR (Yang et al 2021).
>
> > - Yang, Q., Simão, T. D., Tindemans, S. H., and Spaan, M. T. J. (2021). WCSAC: Worst-case soft actor critic for safety-constrained reinforcement learning. *AAAI*, 10639–10646. <https://doi.org/10.1609/aaai.v35i12.17272>
>
> - **W8.** _The environments tested share similar underlying physics._
>
> Please refer to our global response **General Comment 4** for clarifications about this point.
>
> - **W9.** _The paper's treatment of model parameterization through simple mass and friction variations inadequately captures the complexity of real-world dynamic variations._
>
> While we agree that the target environments may not be sufficient for real-world dynamics variations, we would like to argue that such environments are already sufficient to show that a naive approach would fail to perform a safe transfer.
> Our work shows that it is possible to robustify a safe policy and safely transfer it to new environments.
> We believe this will spark the study of more realistic target environments.
>
> - **W10.** _The Lagrangian multiplier approach for policy imitation lacks theoretical convergence guarantees when combined with robustification techniques, potentially leading to unstable learning dynamics._
>
> Only the student is trained with the Lagrangian policy imitation approach,
> while only the guide is trained with robustification techniques.
> Despite that, it is true that we do not provide theoretical convergence guarantees, but instead have empirical validation,
> which aligns with the experiments by (Yang et al., 2023)
> that display convergence in practical settings.
>
> We hope this clarifies the reviewer's questions. We are happy to follow up if something is not clear.

---

### Official Review · Reviewer_LbED · 2024-11-03

**Soundness:** 2
**Presentation:** 3
**Contribution:** 2
**Rating:** 6
**Confidence:** 3

**Summary:**

The paper investigates robust transfer learning of safety-constrained RL problems. The contribution includes: (1) propose a robust transfer learning framework, (2) theoretical analysis on the safety guarantee in transfer across environments with different dynamics, (3) consolidate safe transfer learning metrics and devise methods to train robust agent.

**Strengths:**

1. The paper is written in an easy-to-understand manner and I appreciate the visual illustration in the paper.

2. The paper did a good background study on robust transfer learning problems and combine methods & metrics from related literature to form their own transfer learning framework.

3. The paper clearly defines the assumptions, mapping, metrics used in their framework and elucidates under what circumstances is safety guaranteed for transfer across tasks with different dynamics.

**Weaknesses:**

1. The paper highlighted that what sets itself apart (from prior work) is that it does not assume uncertainty set is known. However, the uncertainty set is assumed to be bounded by $\epsilon$. I am not too sure if this assumption can be easily verifiable for different set of tasks, especially when $\epsilon$ is related to $\delta$ in theorem 1: when the norm-distance is high, the more we need to "robustify" to have safety assurance. The bound on the norm-distance $\epsilon$ does not seem to be easily quantified.

2. In theorem 1, the paper concludes that $\delta$ should be increased (i.e. making the policy more robust to dynamics that may be dissimilar to the training environment) to have safety assurance. However, this $\delta$ parameter does not appear in subsequent chapters. In particular, Chapter 5 (robust training) does not seem to reconnect with this $\delta$ parameter and the theoretical safety guarantee. The theory and the actual training algorithm seems a little disjoint in this regard.

3. Expanding on item 2 above, how is the problem formulation in "formal statement of the problem" connected to the parameter $\delta$? I am also not too sure how does the training algorithm weigh the noise / perturbations (in Entropy Maximization, Random Action Noise & Adversarial Perturbations) such that sufficient $\delta$ can be reached, in order to achieve good safety performance.

4. The robust training framework depends on the abstraction function $\sigma$. I am not too sure if this abstraction function can be specified easily in practice? Especially when this function needs to connect the dynamics, reward and cost functions (in source and target environments) in accordance to the formulae listed in Definition 2. It may seem straightforward to define $\sigma$ in the empirical analysis section of this paper since source/target envs are largely similar (albeit with different physics parameter) but I guess in practice it may not be that easy.

5. Chapter 5.1 introduces state abstraction function and I guess this requires substantial feature engineering effort? Is this exploration bonus a bit short-sighted since it only computes the norm-distance between the current state and the next state?

6. Chapter 5.2 needs to be expanded: the paper mentions that a Lagrangian multiplier $\lambda$ is used to weigh similarity between two policies. I can't figure out how would this enable "student's policy imitate the guide's whenever the cumulative cost is above the safety threshold". In addition, how does the algorithm decide the correct $\lambda$ value to apply during training? These should be further discussed.

7. The safety gymnasium task analyzed in empirical analysis seems to be only Goal task. Having more tasks may strengthen the paper in this aspect.

**Questions:**

- Please refer to the bulleted items listed in the Weaknesses section. I'm particularly curious about how does the algorithm quantify and increase $\delta$ to achieve acceptable level of safety.

- Also, in the last paragraph of Section 6.2, the authors mentioned that "values shown in the tables are the expected costs of one batch of guides". Is there any challenge in getting the expected costs from multiples batches?

- Is there any reason why the students are limited to off-policy RL algorithms: SAC and DDPG? For deterministic policy, why is DDPG tested over TD3?

- In Appendix F (Safety Jump-Start Heatmaps), non-deterministic policy seems to be less robust than deterministic policy. Could you discuss why? This seems counterintuitive to me, especially for a CMDP task.

- I'm more than happy to discuss and please let me know if there's anything I misunderstood or missed out.

---

> ### Author Response · Authors · 2024-11-23
> **Response to Reviewer LbED (1)**
>
> First, we would like to thank reviewer LbED for the insightful comments and openness for further discussion. Your comments have helped us improve the overall quality and clarity of the paper.
>
> Following the reviewer's **(Q)uestion1**, we start addressing the weaknesses pointed out by the reviewer and then address the remaining questions.
>
> - **(W)eakness1.** _The paper does not assume the uncertainty set is known. However, the uncertainty set is assumed to be bounded by $\delta$._
>
> Boundedness helps us provide guarantees, as it does not allow dynamics functions
> within the uncertainty set to be infinitely different from each other.
> While the uncertainty set is bounded by some $\delta$ by assumption, it is crucial to note that we do not know this value during training.
> Theorem 1 serves as a "sanity check", to verify that there exists a ball of radius $\delta$ such that if the agent is robustly safe, then it will also be safe after the transfer.
> It shows that there is a good reason to compute a robust policy, that is, to increase $\delta$.
> Lastly, similar assumptions are relatively common in this kind of setting. For example, Eysenbach and Levine (2022) show that robustness achieved through entropy maximization results in a policy that is safe within a ball of radius $\epsilon > 0$ around the nominal dynamics.
>
> > - Eysenbach, B. and Levine, S. (2022). Maximum entropy RL (provably) solves some robust RL problems. In ICLR. OpenReview.net.
>
> - **W2 and W3.** _The parameter $\delta$ presented in section 4 does not appear later on. Connection between the level of noise/perturbations ($\alpha$) such that sufficient $\delta$ can be reached._
>
> This is a fair criticism; thank you for bringing it up. Please refer to our global response **General Comment 3** for clarifications about this point.
>
> - **W4.** _The robust training framework depends on the abstraction function $\sigma$. I am not too sure if this abstraction function can be specified easily in practice._
>
> Please refer to our global response **General Comment 1** for clarifications about this point.
>
>
> - **W5.** _State abstraction function for exploration requires substantial feature engineering effort and exploration bonus is limmited_
>
> We agree with the reviewer regarding the feature engineering for exploration. The current reward bonus based on the distance of a subset of the features only induces a limited degree of exploration. The design of more sophisticated model-free exploration to increase the coverage of the state space is still an open question. As this is outside our work's scope, we decided to use the same strategy as Yang et al. (2023).
>
> - **W6.** _Lagrangian multiplier $\lambda$ is used to weigh similarity between two policies. How would this enable the "student's policy to imitate the guide's whenever the cumulative cost is above the safety threshold"._
>
> The Lagrangian multiplier $\lambda$ results from the Lagrangian relaxation of the CMDP objective (Altman et al., 2019). In practice, we fit $\lambda$ via gradient descent. If the student's cost is high, $\lambda$ will increase to incentivize the student to accumulate less cost.
>
> We can then use the same weight $\lambda$ in the strength of the policy distillation.
> In other words, if the student violates the safety constraints, we incentivize them to imitate the guide.
> For a more detailed explanation, we refer to Yang et al. (2023).
>
> - **W7.** _The safety gymnasium task analyzed in empirical analysis seems to be only Goal task. Having more tasks may strengthen the paper in this aspect._
>
> Please refer to our global response **General Comment 4** for clarifications about this point.
>
>
> - **Q2.** _Is there any challenge in getting the expected costs from multiple batches?_
>
> We chose to provide an empirical evaluation of the different guides and vary the type of perturbation, noise level, and environment.
> Evaluating each of these agents in a set of target environments to test their robustness further increases the computational costs.
> In summary, training and evaluating so many configurations already had a high computational cost, and running multiple repetitions was simply infeasible.

---

> ### Author Response · Authors · 2024-11-23
> **Response to Reviewer LbED (2)**
>
> - **Q3.** _Is there any reason why the students are limited to off-policy RL algorithms: SAC and DDPG? For deterministic policy, why is DDPG tested over TD3?_
>
> We test off-policy methods as 1. it allows training the student with the samples collected from the guide 2. they are particularly suitable for safety-critical applications due to sample efficiency.
> Moreover, we specifically use DDPG and SAC as prior work has succeeded with these methods (Tessler et al., 2019; Yang et al., 2023).
>
> > - Tessler, C., Efroni, Y., and Mannor, S. (2019). Action robust reinforcement learning and applications in continuous control. In ICML, volume 97 of Proceedings of Machine Learning Research, pages 6215–6224. PMLR.
> > - Yang, Q., Simao, T. D., Jansen, N., Tindemans, S. H., and Spaan, M. T. J. (2023). Reinforcement learning by guided safe exploration. In ECAI, volume 372 of Frontiers in Artificial Intelligence and Applications, pages 2858–2865. IOS Press.
>
> - **Q4.** _In Appendix F (Safety Jump-Start Heatmaps), non-deterministic policy seems to be less robust than deterministic policy. Could you discuss why? This seems counterintuitive to me, especially for a CMDP task._
>
> We believe that this is due to the "key insight" in section 4.2:  Since the nondeterministic guide is more erratic, it cannot recover from the unsafe state as effectively as the deterministic guides.
>
> We hope this clarifies the reviewer's questions. We are happy to follow up if something is not clear.

---

> ### Comment · Reviewer_LbED · 2024-12-03
>
> I thank the authors for the detailed response and revised manuscript. With most of the review comments clarified, I'm willing to increase my score (note: score has already been updated in my review).
>
> With that said, I still think that (1) expanding the experimented task beyond Goal task and (2) explain why TD3 is preferred over DDPG in experiments, especially when TD3 is a newer (and more stable) off-policy deterministic policy algorithm. I wish these can be addressed in the camera-ready version.

---

> > ### Author Response · Authors · 2024-12-03
> > **Response to Reviewer LbED**
> >
> > We would like to thank the reviewer for the response and for increasing the score towards acceptance.
> >
> > (1) While we believe that evaluating on many target environments---as we did in the paper---is the main way to show the robustness of the method, we acknowledge that testing it in other tasks will reinforce such claims. Therefore, we will evaluate our method on a new task from the Safety-Gymnasium benchmarks for the camera-ready version.
> >
> > (2) We will further discuss the choice of off-policy RL algorithm:
> >
> > > We opt for off-policy methods because they enable training the student using samples collected by the guide, and their sample efficiency makes them particularly suitable for safety-critical applications. Specifically, we employ DDPG and SAC as prior work has shown success with these methods (Tessler et al., 2019; Yang et al., 2023). Nevertheless, other methods have also shown promising results, such as TD3's improved stability over DDPG (Fujimoto et al., 2018). Extending our method to other off-policy algorithms should be relatively straightforward.
> >
> > > * Fujimoto, S., Hoof, H. van, and Meger, D. (2018). Addressing function approximation error in actor-critic methods. *ICML*, *80*, 1582–1591. <http://proceedings.mlr.press/v80/fujimoto18a.html>
> > > * Tessler, C., Efroni, Y., and Mannor, S. (2019). Action robust reinforcement learning and applications in continuous control. In ICML, volume 97 of Proceedings of Machine Learning Research, pages 6215–6224. PMLR.
> > > * Yang, Q., Simao, T. D., Jansen, N., Tindemans, S. H., and Spaan, M. T. J. (2023). Reinforcement learning by guided safe exploration. In ECAI, volume 372 of Frontiers in Artificial Intelligence and Applications, pages 2858–2865. IOS Press.

---

### Official Review · Reviewer_YHLY · 2024-11-04

**Soundness:** 2
**Presentation:** 2
**Contribution:** 3
**Rating:** 6
**Confidence:** 4

**Summary:**

This work seeks to ensure robustly safe transfer learning when the dynamics of a new environment differ from the original environment. The proposed approach (1) strengthens an agent’s robustness within a controlled environment and (2) provides a provable guarantee under some assumptions of safe transfer to new environments. Empirical results show that this method maintains safety after transitioning to a new environment. The authors also propose several metric for safe robust transfer learning, which might be useful.

**Strengths:**

1. The topic is interesting and significant, and the solution is overall reasonable, the presentation of empirical results is clear and easy to understand.

2. The results are mostly technically correct and bring both theoretical novelty and practical insights.

**Weaknesses:**

1. The assumptions are not as mild as claimed by the authors, and there are also many hidden assumptions. For example, the paper requests the existence of a robustly safe policy for under constraints only on $\|P-P'\|$.

2. The technical part is confusing. The authors first claim to solve CMDP but later focus on satisfying constraints on every state-action pair $Q^c(s, a)\leq d$, which is a much more conservative objective. The definition of norm and abstract operator is also sloppy.

3. The robustness heavily relies on the action disturbance. There are many other dynamics-perturbation-based robust MDP algorithm such as [1], the authors should consider additional algorithms.

[1] Robust markov decision processes: Beyond rectangularity
V Goyal, J Grand-Clement - Mathematics of Operations Research, 2023

4. The humans knowledge is important in the proposed formulation and is not properly discussed. For example, the abstraction needs to be safety-irrelevant, but sometimes all the observations are related to safety, and sometimes one does not know which observation is safety-irrelevant.

**Questions:**

1. How is the theorem 1 derived? More specifically, there is a hidden assumption that there exists a $\pi$ that is safe for all $P$ with $\left\|P-P^{\diamond}\right\| \leq \delta$? Solving a robustly safe policy is also a very challenging task, especially for a very large number $\delta$.

2.  Is $P$ a matrix or a nonlinear transition operator? is the state and action space finite or continuous?

3. In the experimental section, why is the mapping from friction and mass to transition dynamics continuous and bounded? For example, in some cases, the friction changes from static friction to kinetic friction, and the change point is usually not continuous.

4. What exactly are the safety criteria are you considering? in the CMDP problem formulation in line 118, you said the goal is $C(\pi)\leq d$, while later you convert it to the constraints on the $Q^c$ function. They are not equivalent and the $Q^c$ constraints are much more restrictive than the $C(\pi)$ constraints.

Other minor questions related to the notations:

1. What is the superscript ${}^*$ means in line 244?

I will consider raising my score if the question are properly answered.

---

> ### Author Response · Authors · 2024-11-23
> **Response to Reviewer YHLY**
>
> We are grateful to reviewer YHLY for their detailed and thoughtful feedback, which has provided us with new perspectives and valuable guidance for improving the paper.
>
>
> - **(W)eakness1 and (Q)uestion1. The assumptions are not as mild as claimed, and there is a hidden assumption that exists a robustly safe policy $\pi$, meaning $\pi$ is safe for all $P$ with $|P - P^\diamond| \leq \delta$.**
>
>
> The reviewer brings up a relevant concern. Please refer to our global response **General Comment 2** for clarifications about this point.
>
> - **W2 and Q4.** _What exactly are the safety criteria are you considering? $Q_c$ contraints are much more restrictive than the $C(\pi)$ constraints, what exactly are the safety criteria? Also, the norm and abstract operator are defined sloppily._
>
> Since most algorithms apply the constraints on $Q_c$ in practice, we also developed our analysis based on $Q_c$.
> This indeed implies that the constraints on the expected cost are also satisfied.
>
> Our theoretical results are independent of the definition of the norm, and one option to measure the distance between two dynamics functions is KL divergence. We will update the paper accordingly.
> Furthermore, we noticed that the abstract operator is defined late in the section; to fix this problem, we will introduce it earlier.
>
>
> - **W3. The authors should consider other perturbation-based algorithms, such as (Goyal and Grand-Clement, 2023).**
>
> Thanks for pointing this out. We will add the discussion to the related work section.
>
> > Overall, planning in robust MDP attempts to find an optimal policy under the worst-case transition of the uncertainty set (Wiesemann et al. 2013), for instance, based on perturbations of the transition function (Goyal and Grand-Clement, 2023). Since we do not have access to the model, we rely on perturbing the input and output of the agent (Moos et al., 2022). In this paper, we consider perturbing the output (action and policy)  of the agent and leave the study of perturbing the inputs (observation, reward, or cost) for future work.
> > - Moos, J., Hansel, K., Abdulsamad, H., Stark, S., Clever, D., and Peters, J. (2022). Robust reinforcement learning: A review of foundations and recent advances. *Mach. Learn. Knowl. Extr.*, *4*(1), 276–315. https://doi.org/10.3390/make4010013
> > - Goyal, V., and Grand-Clément, J. (2023). Robust Markov decision processes: Beyond rectangularity. *Math. Oper. Res.*, *48*(1), 203–226.
> > - Wiesemann, W., Kuhn, D., and Rustem, B. (2013). Robust Markov decision processes. *Mathematics of Operations Research*, *38*(1), 153–183.
>
>
>
> - **W4.** _The abstraction needs to be safety-irrelevant, but sometimes all the observations are related to safety, and sometimes one does not know which observation is safety-irrelevant._
>
>
> Please refer to our global response **General Comment 1** for clarifications about this point.
>
> - **Q2.** _Is $P$ a matrix or a nonlinear transition operator? is the state and action space finite or continuous?_
>
> $P$ is non-linear, and the state and action spaces are continuous.
>
>
> - **Q3.** _Function $u$ maps a mass and friction pair to the corresponding transition dynamics function. Why is this mapping continuous and bounded? This is usually discontinuous._
>
> This function does not need to be continuous, and we will omit the mention of continuity from the paper.
> The function preserves boundedness, that is,
> for any bounded $S \subseteq \mathbb{R}^2$, its image under $u$ is also bounded.
> This implies that $U^\odot$ is bounded in the empirical evaluation, as the mass and friction multipliers range from 0.5 to 1.5.
>
> - **Q5.** _How is superscript * defined (line 244)?_
>
> It is the Kleene closure in this context:  $(S \times A)^* = \bigcup_{n=0}^\infty (S \times A)^n$. We will make this clear in the paper.
>
> We hope this clarifies the reviewer's questions. We are happy to follow up if something is not clear.

---

> > ### Comment · Reviewer_YHLY · 2024-11-26
> >
> > Thanks for replying to my comments. My major concerns have been addressed. I will increase my score to 6. I also urge the authors to thoroughly revise the writing and update the revised manuscript.

---

### Author Response · Authors · 2024-11-23
**Global Response**

Firstly, we thank all the reviewers for their thoughtful comments.
Here, we address common concerns.
We respond to specific questions to each reviewer individually.


- **General Comment 1.** _The abstraction needs to be safety-irrelevant, but sometimes all the observations are related to safety, and sometimes one does not know which observation is safety-irrelevant. Could this abstraction function be specified easily in practice?_

We would like to start this discussion by highlighting that in case the source task's observation space includes all the safety-relevant features from the target task, then it is easy to construct such an abstraction function by stripping away the safety-irrelevant features.
Note that this says nothing about how the dynamics of the two tasks compare; it only describes the observation spaces.
Therefore, the challenge lies in constructing a source environment (such as a simulator) that includes the safety-relevant information from the source task.

Our paper merely assumes that the source environment is provided. Therefore, we focus on training an agent in such a given source (controlled) environment (potentially a simulator).
Afterwards, we transfer the learned policy to the target environment (potentially the real world).
The question of how to obtain the abstraction could be rephrased to how to construct the controlled environment, which is actually orthogonal to our work.
Observe that even if we improve the abstraction function $\sigma$, we still do not expect it to be exact, and we would still require a robustly safe transfer as proposed in this paper.


- **General Comment 2.** _Finding a robust safe policy $\pi$, that is a policy that is safe in all $P$ such that $|P - P^\diamond| \leq \delta$, can be difficult if $\delta$ is large._

This is a challenge in the constrained robust setting when, depending on the problem and the size of $\delta$, finding such a robust, safe policy could be infeasible. However, we would like to point out that:
1. finding such a robustly safe policy can be simple for a significant class of environments and
2. there are alternatives if such a policy does not exist.

We realized these points were not explicit in the submission, so we added the following paragraph to the paper.

> `Note that Theorem 1 does not state that there is a robustly safe policy. If $\delta$ is too large, such a policy may not exist. In practical terms, if we fail to find a feasible policy for a given $\delta$, our method still attempts to find a conservative policy in the source target, which ensures a reduction of safety violations in the target environment. Alternatively, we could search for the largest $\delta' < \delta$ such that a feasible policy exists. Nevertheless, this policy can easily be found in the constrained RL setting. For instance, in problems with an action that does not incur any costs, such as the environment used in the empirical evaluation, the agent can choose this action, which ensures it will satisfy the constraints.'


- **General Comment 3.** _How does perturbation size $\alpha$ relate to $\delta$ and $\varepsilon$?_

To clarify this connection, we will add the following paragraph to the end of Section 5.

> Prior work has studied that perturbation size ($\alpha$) provably makes the policy robust to some $\delta$ (Feng et al., 2020; Eysenbach and Levine, 2022). Therefore, as we increase the value of $\alpha$ we are effectively computing a robust policy for a larger uncertainty set in the source task. Consequently, the agent will be safe in a larger set of target tasks. As a theoretical bound between $\alpha$ and $\delta$ can only be shown in specific scenarios, we adopt an empirical approach in the next section and evaluate the robustness effect across different values of $\alpha$.

> - Feng, F., Wang, R., Yin, W., Du, S. S., and Yang, L. F. (2020). Provably efficient exploration for reinforcement learning using unsupervised learning. In NeurIPS.
> - Eysenbach, B. and Levine, S. (2022). Maximum entropy RL (provably) solves some robust RL problems. In ICLR. OpenReview.net.

- **General Comment 4.** _The environments aren't varied enough._

While we agree that considering other tasks would strengthen our work, we would like to highlight that we considered a large number of perturbations of the dynamics of the agent and the target environment.
This allowed us to provide an extensive overview of the robustness of the proposed algorithm, which is the main aspect of our work.

- **General Comment 5.** _Given that the source code is not attached, reproducibility is low._

We will release the source code on GitHub upon publication.
For now, we have created an anonymized repository to provide the code to the reviewers:

[https://anonymous.4open.science/r/KElzutXbe97jbuT/](https://anonymous.4open.science/r/KElzutXbe97jbuT/)

---

### Author Response · Authors · 2024-11-26
**Revised Submission**

We have just uploaded a revised version of our submission, with all changes marked in red and labeled "NEW" in the margin on the right. Once again, we sincerely thank all reviewers for their constructive feedback, which has greatly contributed to improving the paper. We are more than happy to address any remaining questions or make further improvements as needed.

---

### Meta-Review · Area_Chair_Lrax · 2024-12-23

**Metareview:**

The paper presents a method for robust transfer of safe policies to a perturbed domain with a theoretical analysis of the transfer's safety guarantees. This addresses a very important topic in RL applications and significantly advances our understanding of this area. Reviewers have raised multiple concerns, including regarding clarity, limitations, positioning in existing literature, requirement of domain expertise, soundness, and evaluation. However, these concerns have all reportedly been addressed in author feedback.

**Additional Comments On Reviewer Discussion:**

Reviewers provided a thorough criticism of the paper and raised multiple concerns in all aspects. The authors addressed all of these concerned in their rebuttal, prompting all reviewers to acknowledge their satisfaction with the response and most to raise their scores.

---

### Decision · Program_Chairs · 2025-01-22

Accept (Poster)